# Analysis of changes in the occurrence of ice phenomena in upland and mountain rivers of Poland

**Krzysztof Kochanek**[1]*, **Agnieszka Rutkowska**[2], **Katarzyna Baran-Gurgul**[3], **Iwona Kuptel-Markiewicz**[4], **Dorota Mirosław-Świątek**[5], **Mateusz Grygoruk**[5]

1 Faculty of Building Services, Hydro and Environmental Engineering, Warsaw University of Technology, Warsaw, Poland, 2 Faculty of Environmental Engineering and Land Surveying, Department of Applied Mathematics, University of Agriculture in Cracow, Cracow, Poland, 3 Faculty of Environmental Engineering and Energy, Department of Geoengineering and Water Resources Management, Cracow University of Technology, Cracow, Poland, 4 Department of Hydrology and Hydrodynamics, Institute of Geophysics Polish Academy of Sciences, Warsaw, Poland, 5 Department of Hydrology, Institute of Environmental Engineering, Meteorology and Water Management, Warsaw University of Life Sciences-SGGW, Warsaw, Poland

* Krzysztof.Kochanek@pw.edu.pl

**Data Availability Statement:** All relevant data are within the manuscript and its Supporting information files.

## Abstract

The ice phenomena are an inherent component of rivers in temperate, continental, and polar climate zones. Evident progress in global warming leads to a decrease in snow cover on land and ice phenomena in water bodies, disrupting the stability of the hydrological cycle and aquatic ecosystems. Although common observations indicate the disappearance of ice phenomena in rivers over recent decades, detailed quantitative research is lacking in many regions, especially in the temperate zone. In this paper, ice phenomena were analyzed on the rivers of southern Poland, located in the upland and mountain areas of the country, as no such studies have been conducted so far. The temporal changes in the annual number of days with ice (NDI) phenomena were studied in locations where ice phenomena were observed every year for at least 30 years between 1951 and 2021. Using straightforward but commonly accepted procedures, such as the Mann-Kendall test, statistically significant decreasing trends in the annual NDI were revealed for the majority of gauging stations. The Theil-Sen (TS) slope mean values were -1.66 (ranging from -3.72 to -0.56), -1.41 (from -3.22 to -0.29), and -1.33 (from -2.85 to -0.29) for the datasets representing the periods 1992–2020, 1987–2020, and 1982–2020, respectively. The results for the annual NDI were additionally presented within the context of meteorological characteristics such as annual and winter (Nov-Apr) air temperature, precipitation, and water temperature. Correlation and regression analyses revealed that the main factor triggering the decrease in NDI is the increase in the average winter air temperature. An increase in temperature by 1°C results in a decrease in NDI by up to twenty days. If these negative trends continue, ice phenomena may disappear completely from southern Polish rivers within few decades.

**Funding:** The author(s) received no specific funding for this work.

**Competing interests:** The authors have declared that no competing interests exist.

## Introduction

Global climate change poses severe threats to the environment and economies [1]. Beyond the intuitive and widely discussed consequences of global warming, such as temperature rise, changes in precipitation patterns, and related flow regime alterations, the ice phenomena (namely: grease ice, floe, shore ice, ice cover, and ice jams) occurring in surface water bodies within temperate-to-arctic climatic zones are also severely impacted by the changing climate [2–5]. The changes in ice phenomena regarding their duration, occurrence, disappearance, and types correlate with an increase in air temperature and other climatic indices in the region. Most studies revealed climate-induced changes in the duration and spatiotemporal patterns of ice phenomena on rivers, lakes, and coastal waters [6–10]. For example, a study on Russian rivers across Eurasia by Smith [11] found a decrease in ice cover in the 1950s, an increase in the 1980s, and again a decrease in the 1990s, with melt onset shifting negatively by approximately 1 to 3 weeks, consistent with Siberian temperature trends. He also documented interannual and regional variability, such as earlier autumn freezing in Karelian and Kola-region rivers, leading to increased winter ice cover in these areas. Grześ and Ćmielewski [5] confirmed the reduction of days with ice in Arctics (Russian Siberia, Canada, and the USA)–the rivers got frozen later and melt melted earlier in the 1980s when compared to the 1950s, so the number of days with ice was shortened by at least 20 days per 100 years. Contrastingly, Hallerbäck et al. [12] reported a significant reduction in river ice duration across Sweden from the early 20th century to 2021, with the mean number of ice days decreasing by 11 days in the north and 28 days in the south over the last three decades compared to the previous three decades. Similarly, extreme events in recent decades, such as late freezing, early ice melt, and shorter ice cover periods (or no ice cover at all), contribute to the general ice loss in Northern Hemisphere lakes [13]. Klavins et al. [14] noted a reduction in ice cover duration in 17 rivers in the Baltic countries and Belarus based on over 200 years of ice phenomena, hydrological and air temperature observations; in the last several decades the number of days with ice shrunk by 2.8–6.3 days per 10 years, depending on the river.

In Poland, observed and modeled increases in seasonal air temperatures [15] during winter (December to February) underscore the shrinking of winter ice phenomena on rivers observed since the early 19th century. Paczowska [16] reported rises in the percentage of ice-free years: an increase from 10% for the River Warta, 4% in the River Vistula, and 0% in the River Neman from 1822–1877 to 40%, 40% and 30% in the period 1926–1935 in Rivers Warta, Vistula, and Neman, respectively. Simultaneously, the duration of time when ice phenomena were observed during winters shortened significantly. For instance, on the River Vistula, ice cover lasting 81–100 days was observed 26 times from 1878 to 1900 and 35 times from 1901 to 1935, whereas 121–160 days of ice cover was observed 22 times from 1878 to 1900 but only 15 times from 1901 to 1935. This reduction in ice cover duration was echoed across other Polish rivers between 1822 and 1935. Similarly, Pawłowski [17,18] analyzed the Vistula River in Toruń in 1814–2003, noting a decline in ice cover duration from 60–120 days in the 19th century to 30–80 days in the second half of the 20th century, and the duration of any type of ice phenomena decreasing from 88 to 53 days (1882–2011) and ice cover from 40 to 7 days. Pawłowski suggested that these reductions might be partially attributed to river regulation and the construction of the Włocławek reservoir upstream of Toruń, which actually goes along with earlier monumental research by Grześ [19]. Fukś [20] reviewed the impact of dam reservoirs on river ice characteristics in the Northern Hemisphere, identifying climate change as the main cause of later freezing and earlier ice melt, and noting the effects of artificial reservoirs on local river ice phenomena.

Recent studies, such as by Somorowska [21], reported that increased air temperatures (by 2.7˚C over the last 70 years) in central Poland have led to changes in snowfall, rainfall, and baseflow metrics, influencing river flow regime changes in Liwiec River. Additionally, an analysis of ice phenomena duration in several coastal rivers at the southern Baltic from 1956 to 2015 indicated a shortening of ice phenomena occurrence by up to 7 days per decade [8]. For the same region, Łukaszewicz and Graf [22] showed that increasing air temperatures have led to higher river water temperatures, resulting in a shortened duration of ice phenomena.

Bączyk and Suchożebrski [23] analyzed the duration of ice cover in a sequence of gauging stations along the River Bug (east Poland) and found that it shortened from approximately 100 days (1903–1960) to about 60 days (2001–2012) due to higher winter temperatures. The results listed above were the main research impact behind the studies on the temporal changes of the NDI in Polish rivers in this paper.

Ice phenomena and changes in their spatial and temporal patterns affect a wide range of processes and activities. Economically, icing on lakes and rivers challenges winter transportation and navigation. However, in some regions, icing can also enhance transport across rivers and lakes (e.g., in Siberia). Some authors suggest that ice phenomena could be part of cognitive, environmental, specialist, hiking, winter, seasonal, or occasional tourism [24].

However, inland navigation (technical ice breaking) and low water quality hinder comprehensive documentation of changes in ice phenomena over time [25]. Similarly, while studies of rivers impounded by large reservoirs provide important documentation of anthropogenic influence, they often do not reveal the possible influence of climate-related changes in air and water temperatures on ice phenomena [18]. Hydropower operations can extend ice cover duration upstream of hydroelectric facilities, causing severe ice jams and floods, while significantly shortening ice phenomena downstream [26].

Environmentally, changes in ice phenomena duration and patterns affect water temperature inversion and oxygen circulation in aquatic ecosystems [27]. Knoll et al. [28] discussed cultural ecosystem services and benefits related to ice, while Lindenschmidt et al. [29] indicated that prolonged periods without ice can deteriorate water quality. McBean et al. [30] highlighted the depletion of dissolved oxygen due to continued respiration, potentially reducing primary productivity and reaeration.

Under-ice processes impact the annual cycling of energy and carbon through aquatic food webs [31]. Berilsson et al. [32] described how under-ice conditions alter lake physics, affecting auto- and heterotrophic micro-organism distribution and metabolism. Extended periods without ice negatively influence aquatic fauna and flora, extending the growing season for warm-water species and expanding pelagic habitats [33,34]. This extended growing season stresses micro-organisms, favoring competitive species, disrupting life cycles, and altering the reproductive periods of fish and crustaceans [35,36].

Documenting cross-sectoral consequences of changing ice phenomena types, durations, and occurrences is critical. However, extensive, systematic studies covering broad, country-wide data have not yet been conducted in Poland. Despite common perceptions of disappearing ice phenomena on Polish rivers in recent decades, detailed research in the temperate climatic zone is still lacking.

The main objective of this paper is to study changes in the occurrence of ice phenomena in the scarcely documented upland and mountain rivers of Poland, in relation to meteorological conditions. Being aware of the newest methodological approaches for detecting trends in environmental and hydrological variables, such as the innovative trend pivot analysis method and trend polygon star concept method [37], machine learning, and artificial intelligence, here the emphasis was put on the temporal changes in the annual number of days with ice (NDI) using commonly accepted statistical methods, i.e., the Mann-Kendall test with Bonferroni

correction. Note, that this analysis considers all types of ice that may be formed in rivers. Our results remain an input into the riverine ice investigations and aim to contribute to the recognition of ice phenomena alterations using the widest dataset and geographic extent available.

## Materials and methods

### Data

The datasets used for analysis and calculations are based on observations collected by the Institute of Meteorology and Water Management—National Research Institute (IMGW-PIB) (the Official Polish Hydrological Service) and cover the period from 1951 to 2021. The IMGW-PIB repository contains daily observation series of river flow characteristics and ice phenomena from over 1,000 hydrological stations located at various rivers, creeks, and lakes in Poland. The temporal coverage of the particular observation stations varies and rarely covers the entire period of 1951–2021. Nevertheless, such long datasets constitute valuable material for statistical studies and were included in the analysis, regardless of their location in Poland.

Measurements and observations of ice phenomena are carried out every day of their occurrence (except for the measurement of ice cover thickness). The results of the observations are either immediately transferred to the database system or sent to a dedicated application once a month in the form of a log of gauging station observations. The IMGW-PIB's database provides the following information: the code of the station, name of the river or lake, date of the observation (in the hydrological calendar), ice thickness (if observed), code of the ice phenomena (e.g. grease ice, floe, shore ice, ice cover, ice jam, etc.), and the percent of the river's width covered by ice. Gaps in the datasets (sometimes even a few years long) mean that the ice phenomena either were not recorded or noted.

In this work, the annual number of days with ice as the key variable was analyzed. The IMGW-PIB's repository was recently (in 2022) completed for the entire territory of Poland; however, some data are still being reviewed and corrected by the IMGW-PIB and are currently unavailable to the research community.

While analyzing the datasets, we noticed significant errors in the data for central and northern Poland, such as the notation of ice phenomena in the middle of summer (which is impossible in Poland). These flawed datasets would invalidate the results and conclusions drawn from the entire set of available information. Therefore, we analyzed each dataset thoroughly to eliminate potentially invalid data and to select only high-quality and reliable data.

The selection of the series was based on the assumption that at least one day with any type of ice phenomena was observed each year for 30, 35, 40, or 50 years. Only sequences with a non-zero number of days with ice phenomena each year were considered. Years with no ice in any of its forms were excluded. As a result, out of 1,000 available datasets, we employed the gauging stations mostly located in the southern part of the country. This region accounted for the completion of the ice phenomena analysis in Poland, as previous research concentrated on the central and northern parts of the country, where most lowland catchments are located.

The selection process was conducted in a two-step approach:

Step 1: Selecting datasets that end in 2021 and cover consecutive years.

   Class 1–30-year series (from 1992): 40 stations
   Class 2–35-year series (from 1987): 39 stations
   Class 3–40-year series (from 1982): 37 stations
   Class 4–50-year series (from 1972): 1 station
   Note: All 37 gauging stations of the 3rd class fulfil the criteria of the 39 and 40 stations of

the 2nd and 1st classes, respectively. In other words, a 40-year data series belonging to the 3rd class also has a 35-year subseries in the 2nd class and a 30-year subseries in the 1st class.

Step 2: Selecting the longest available data sequences (longer than 50 years) with at least one day of ice per year from the entire study period (1951–2021), regardless of the starting year. This step resulted in 5 data series. One of these five series ends in 2021, so its 30, 35, 40, and 50-year subseries also belong to Step 1 classes.

The location of the analyzed gauging cross-sections, the size of the catchments, and the histograms of the catchment area are illustrated in Fig 1. The characteristics of the selected gauging cross-sections and the catchments enclosed by those sections are listed Table A1 in S1 Appendix.

An additional crucial factor in the occurrence of ice phenomena is the elevation of the gauging station. Upland and mountain rivers (in the south of Poland; see Fig 2) are more prone to experience ice phenomena due to higher elevations and lower temperatures compared to lowland rivers in the central, western, or northern parts of the country. The considered gauging stations represent mostly medium-sized upland and mountainous catchments in Poland (Figs 1 and 2).

- 25 gauging stations enclose catchments of areas between 100 and 1,000 km$^2$.

- 11 stations belong to the lowland regime and are located below 300 m a.s.l.

- 9 stations are located in high mountains at altitudes above 500 m a.s.l.

- 5 gauging stations with the longest data sequences (minimum 50 years without gaps) are located within the Vistula River catchment and are relatively evenly distributed along its course (Fig 3 and Table 1).

Since air temperature is considered a primary factor determining the form and duration of ice phenomena on rivers, this study also examines long-term trends in the average annual and average winter (Nov-Apr) temperatures at synoptic stations. Nine synoptic stations from the IMGW-PIB database, located in proximity to the analyzed river gauging cross-sections, were selected. The average distance from the gauging cross-sections to the nearest synoptic stations is only 26.2 km, thus the temperatures at the nearest synoptic station can represent the temperature at the gauging cross-section.

The synoptic stations assigned to the gauging cross-sections are listed Table A1 in S1 Appendix, and their locations are shown in Fig 1. The temperature characteristics were derived from daily mean air temperature series. The average annual and average winter temperatures for the period 1992–2021 are provided Table A1 in S1 Appendix. They vary from 6.3 to 9.4˚C (annual scale) and from 0.1 to 3˚C (winter), with the lowest temperatures recorded in the mountains. The spatial average temperature is 8.5˚C (annual) and 2.1˚C (winter).

Additionally, the following characteristics were studied: the seasonal sum of precipitation (annual, winter) and the maximum daily precipitation (annual, winter). The data were obtained from the synoptic stations. Water temperature characteristics, such as the average annual and average winter temperatures, were analyzed in a comparative study with air temperatures. Although water temperature measurement records are scarce in the hydrological database analyzed, three water level gauging stations with complete data records (1992–2021) were identified, enabling such an analysis. Additionally, trend analysis in the number of days with ice was conducted at the stations with the longest data records (Table 1). Similarly, the synoptic stations were assigned to the stations from Table 1, and their locations are shown in

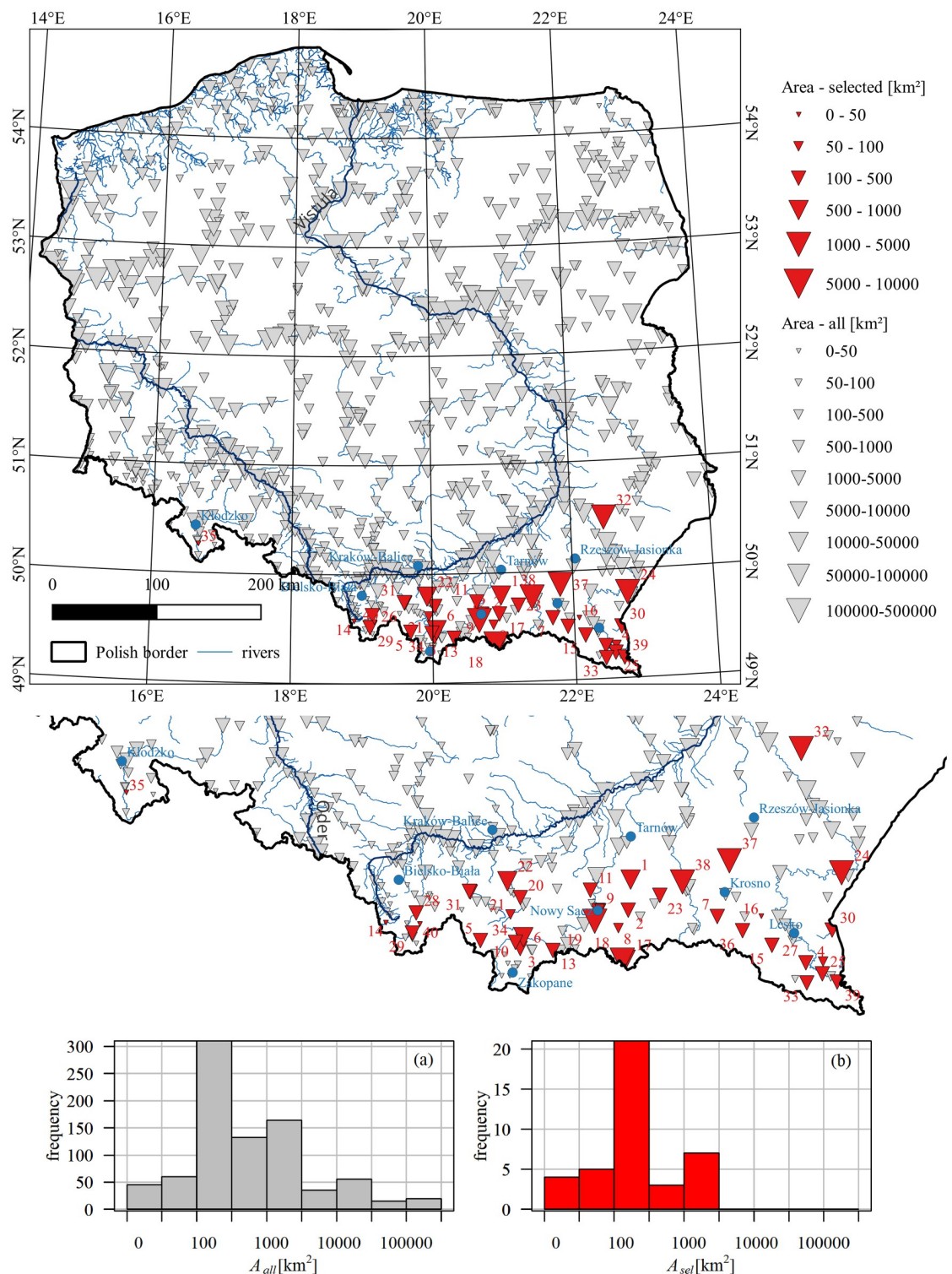

**Fig 1.** The spatial distribution of the selected 40 cross-sections (red triangles) amongst all cross-sections (grey triangles) across Poland from the IMGW-PIB database; the triangle size reflects the catchment area and the histograms of the catchment area within individual classes: a) grey–all stations ($A_{all}$), b) red–selected stations ($A_{sel}$). Blue points on the maps show the locations of the nearest synoptic stations assigned to the cross-sections selected for the study.

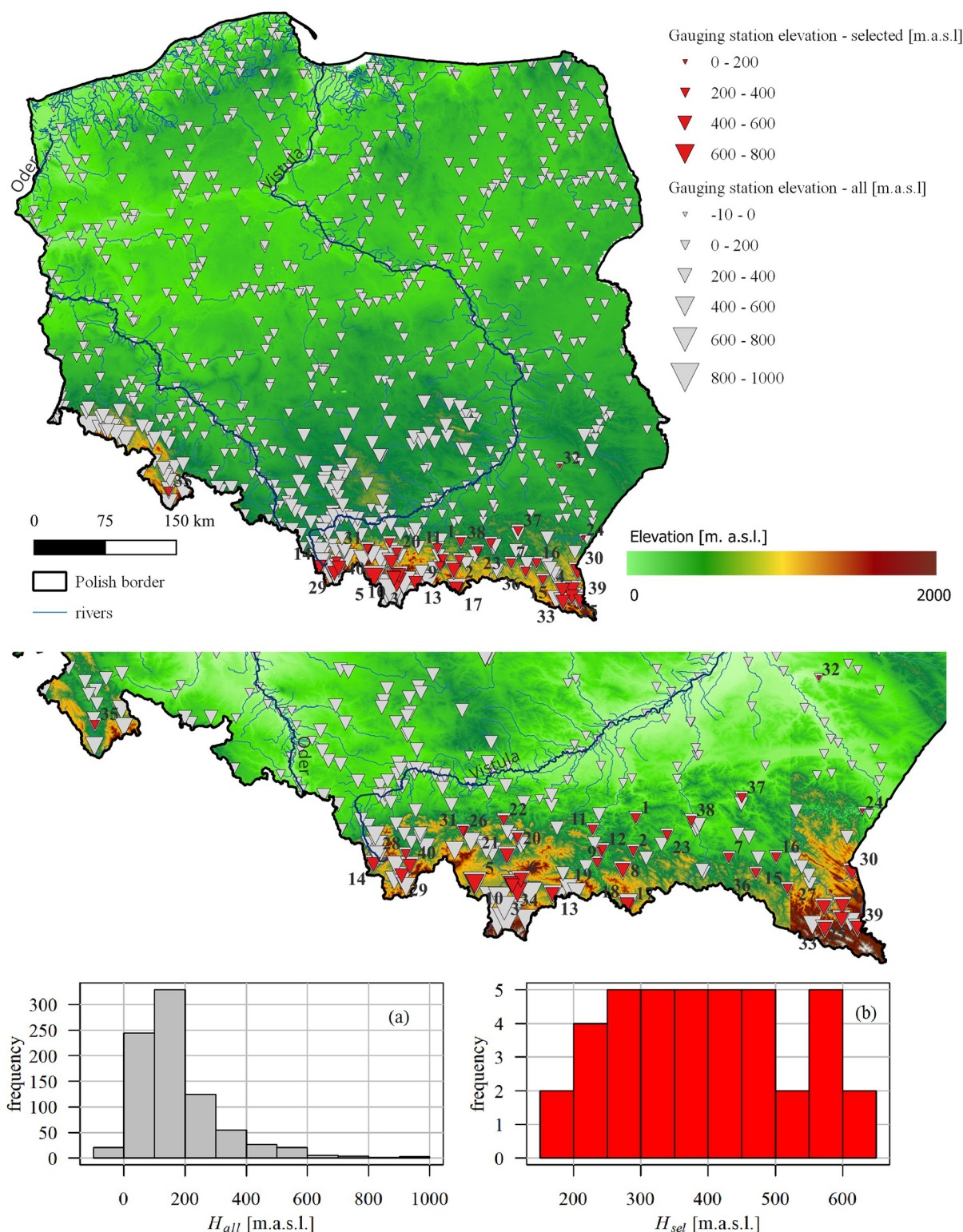

**Fig 2. The selected cross-sections (red triangles) and all cross-sections (grey triangles); the triangle size reflects the cross-section's altitude.** The numbers on the maps at the red triangles correspond with the numbers of the gauging stations listed Table A1 in S1 Appendix. The histograms of the altitude (grey and red–all and selected cross-sections, respectively) show that the higher-elevated cross-sections dominate among the selected ones. The hypsometric map (DEM) is available at: https://www.geoportal.gov.pl/pl/usluga/uslugi-przegladania-wms-i-wmts (accessed on 3 April 2024).

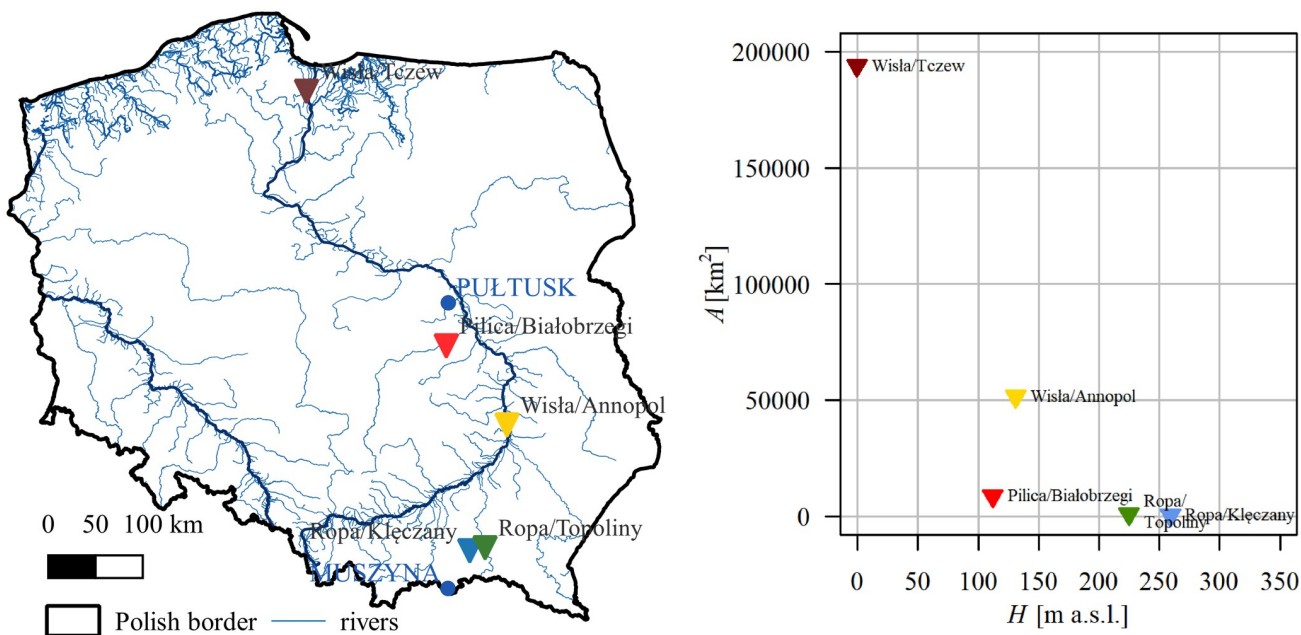

**Fig 3. The characteristics (location, area, and altitude) of the 5 gauging stations with the longest sequences and at least one day with an ice phenomenon per year within the study period selected from the complete study period (1951–2021).** Blue circles on the map show the location of the synoptic stations.

Fig 3. To explain the causes of the changes in the number of days with ice over the long period, trend analysis of the average annual and average winter air temperatures was carried out.

The data availability, annual number of days with ice, and completeness of the datasets, which determined the choice of gauges for the analysis, led to uneven coverage of the country with measurement points. The selected gauging stations are mostly located in small catchments in the Carpathian Mountains, in the southern part of Poland, leaving vast areas beyond our investigation (compare red and grey triangles in the maps in Figs 1 and 2). This is mainly due to the fact that only stations with complete, correct datasets and ice observations every year for the last 30, 35, 40, or 50 years were selected for this analysis. As revealed by Huh et al. [38], 30 years of observations are regarded as a sufficient length of data series to responsibly describe temporal trends of selected hydrological variables used to characterize flow regimes.

Being located in the temperate climate zone [39], Poland and the Western Outer Carpathian Mountains (medium-sized mountains, the highest peak is Gerlach at 2655 m a.s.l.) are characterized by a humid maritime climate in the western part and a dry continental climate in the eastern part. The average January temperatures range from -2ºC to -4ºC, while in July

**Table 1. The gauging stations with the longest sequences with at least one day with an ice phenomenon per year within the study period.**

| Years | Period | River | Gauging cross-section | Area | Altitude | GPS Coordinates | |
|---|---|---|---|---|---|---|---|
| | | | | $A$ [km$^2$] | $H$ [m a.s.l] | Longtitude | Latitude |
| 62 | 1960–2021 | Ropa | Klęczany | 484.1 | 259.1 | 21.217 | 49.701 |
| 60 | 1960–2019 | Ropa | Topoliny | 974.2 | 224.8 | 21.444 | 49.725 |
| 67 | 1951–2017 | Vistula (Wisła) | Annopol | 51515.2 | 130.9 | 21.834 | 50.886 |
| 52 | 1968–2019 | Pilica | Białobrzegi | 8664.6 | 112.0 | 20.952 | 51.657 |
| 60 | 1951–2010 | Vistula (Wisła) | Tczew | 193922.9 | -0.56 | 18.802 | 54.096 |

they range from +18ºC to +21ºC [40], which guarantees climatic variability conducive to creating ice phenomena in the rivers. The selected stations represent a spectrum of the hydrological regimes of rivers in Poland, including fast-flowing mountain rivers such as the Dunajec River and large lowland rivers such as the Vistula (Wisła) River. The results of the study constitute a valuable contribution to knowledge about ice occurrence in upland and mountainous catchments.

## Methodology

**Relationships between catchment characteristics and ice occurrence.** To determine whether the frequency of ice occurrence, represented by $N_{mean}$–the annual mean number of days with ice, can be dependent on catchment characteristics such as $H$–gauging cross-section altitude, or $A$–catchment area, the Spearman correlation coefficient $r_s$ between $H$ and $N_{mean}$ and between $A$ and $N_{mean}$ was computed for a sample of 40 gauging cross-sections with 30 years of data. Next, the hypothesis that the coefficient is significantly different from zero was verified. A similar analysis was conducted for station altitude $H$ and catchment area $A$ to recognize the relationship in catchments where ice was observed each year.

**Temporal variability in the annual sum of days with ice and temperature characteristics.** From the perspective of assessing how climatic or human-induced changes may impact river ice occurrence, the temporal variability of $X_t$ was examined. Here, $X_t$ represents the annual sum of days with ice or a climatic characteristic, including the mean temperature (annual, winter), total precipitation (annual, winter), maximum daily precipitation total (annual, winter), and average/mean winter water temperature. The null hypothesis was tested regarding no trend in $X_t$ as $t$ increases, in contrast to the alternative hypothesis suggesting a decreasing trend. To test the null hypothesis, the Mann-Kendall (MK) test was utilized [41,42]. This non-parametric method was chosen due to the non-normal distribution of $X_t$ at most stations, which was confirmed using the Shapiro-Wilk test [43]. Two versions of the MK test were applied: the first version for series where autocorrelation at lag = 1 year was assumed to be zero, and the second version for series with autocorrelation significantly different from zero at lag = 1 year. Both versions were considered to address the impact of autocorrelation on the test's accuracy in detecting trends and drawing conclusions. Initial data analysis was performed to determine lag values with substantial autocorrelation. Ultimately, only the version with lag = 1 year was utilized, as no significant autocorrelation was observed at longer lags. A significance level of $\alpha = 0.05$ was applied in the study.

Assuming that $x_1, x_2, \ldots, x_n$ are sample values of the variables $X_t$ at $t = 1, 2, \ldots, n$, the formulas of the MK test statistic in the first version are:

$$S = \sum_{i=1}^{n-1} \sum_{j=i+1}^{n} \mathrm{sgn}(x_j - x_i), \ \mathrm{var}(S) = \frac{n(n-1)(2n+5)}{18} \tag{1}$$

where sgn is the signum function. If the sample length $n$ is greater than 10 then the variable

$$Z_{MK} = \frac{S - \mathrm{sgn}(S)}{\sqrt{\mathrm{var}(S)}} \tag{2}$$

follows the standardized normal distribution, $N(0,1)$. The left-sided $p$-value of the test is obtained from this.

In the second version, if $\rho_1$ –the value of the autocorrelation function at lag = 1 year is significantly different from zero, then the actual variance is underestimated in Eq (1). Hence the

correction factor should be used for $var(S)$, namely [44]:

$$\text{var}^* S = \text{var}(S)\frac{n}{n^*}, \tag{3}$$

where $n^*$ is the effective sample size and:

$$\frac{n}{n_S^*} = 1 + \frac{2}{n(n-1)(n-2)}\sum_{i=1}^{n-1}(n-i)(n-i-1)(n-i-2)\rho_s(i), \tag{4}$$

where $\rho_s(i)$ is the lag $i$ autocorrelation function of the ranks of the observations. The $Z_{MK}$ statistic is then computed from Eq (2) with $var^*S$ in the denominator.

**Controlling the false discovery rate.** As multiple MK tests were conducted simultaneously, it was necessary to control the false positive rate. To achieve this, the Bonferroni correction was employed [45] to ensure that the probability of having at least one false positive was less than the level of significance ($\alpha = 0.05$). This procedure was chosen due to its robustness in handling dependencies between sites [46].

**The Theil-Sen slope estimator.** In the cross-sections where the trend was identified, the Theil-Sen estimator of the slope of the regression line was computed [47,48],

$$\beta = median\left(\frac{x_j - x_i}{j - i}\right), \text{ for } j > i \text{ where } i, j = 1, 2, \ldots, n \tag{5}$$

The slope value shows a rate of change and is an indicator of the strength of the decreasing/increasing tendency because the higher its absolute value, the stronger the changes in the number of days with ice. The estimator is often used if the traditional slope coefficient in the linear regression model is less reliable because the normality of the distribution of $X_t$ cannot be assumed.

**Correlation and regression analysis.** The Pearson correlation coefficient between the number of days with ice and the mean winter temperature from the nearest synoptic station was derived. The hypothesis was verified that the two variables are negatively correlated. The linear regression model $X = aT + b + \varepsilon$, where $X$ is the NDI (the target variable), $T$ is the mean winter temperature (the explanatory variable), and $\varepsilon$ are residuals, was designed.

## Results

### Trend analysis in the number of days with ice and temperature

For the studied gauging stations over the most recent 30 years (1992–2021), the average of the mean number of days with ice phenomena per year ($N_{mean}$) is 55.5 days. This average varies from 30 to 81 for individual stations (refer to Fig 4). In the majority of stations, the average number of days with ice phenomena per year ranges approximately from 40 to 50 during the studied period. Few stations had an average number of days with ice phenomena exceeding 70 days. It is notable that gauges with lower mean number of days with ice are predominantly situated in the south, while those with higher $N_{mean}$ are in the northern part of the Carpathian study area.

The Spearman correlation coefficient between the altitude of the gauging stations and $N_{mean}$ (the mean number of days with ice phenomena per year) is found to be $r_s = 0.50$ across all 40 stations. A significant positive correlation exists between these two variables, as indicated by the hypothesis testing results showing a $p$-value of 0.00104. This suggests that as the gauging stations' elevation increases, the duration of icing periods also tends to increase.

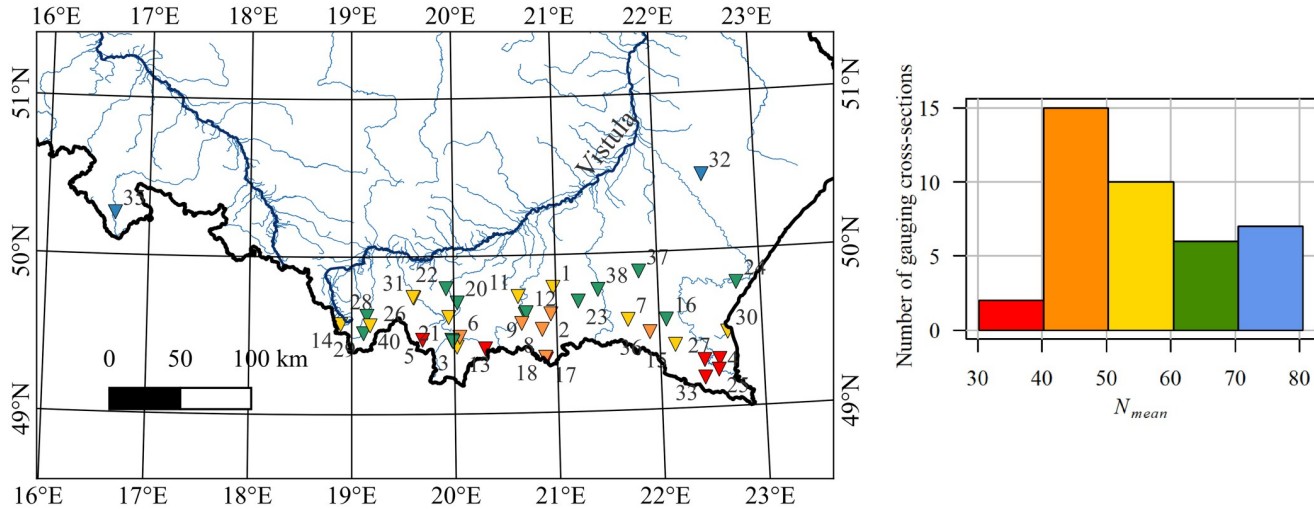

**Fig 4. The spatial distribution of the gauging cross-sections and the histogram of $N_{mean}$—The mean number of days with ice per year (1992–2021).** The colours on the map correspond with the colours in the histogram.

The analyses were conducted for series spanning 30, 35, 40, 50, and over 50 years of ice data since any shorter data series may compromise statistical analyses. The rarer, lengthiest datasets, which encompass over 50 years of ice data, are considered as a benchmark for comparison with other data sets.

The statistically significant trends (indicated by arrows in Table 2) in the annual sum of days with ice suggest a decline in ice cover in southern Polish rivers across the majority of cross-sections, irrespective of their location and river size/altitude. This trend is evident in the summary presented in Table 3, where statistically significant decreasing trends prevail across all cross-sections. This declining trend was observed in 73%, 67%, 81%, and 100% of the cross-sections with data series of lengths 30, 35, 40, and 50 years, respectively. In the sole 50-year data series for the Klęczany station at the Ropa River, the $Z_{MK}$ value is -1.98, with a Theil-Sen slope of -0.46. Some stations did not display any discernible trend (represented by dots in Table 2). Interestingly, no stations showed positive trends.

The trend in series of different lengths showed consistent direction; however, the magnitude, represented by the Theil-Sen slope value, can vary significantly. The absolute values of the Theil-Sen slopes were generally noticeably higher for shorter datasets compared to longer ones within the same cross-sections, particularly evident when comparing 30 years to 35 years. This suggests that the most significant decline in the annual number of days with ice has occurred in the past three decades as depicted in Fig 5.

For instance, in cross-sections exhibiting a decreasing trend in ice occurrences, the Theil-Sen slope ranged from -3.72 to -0.56 days/year for 30 years of ice data, from -3.22 to -0.29 days/year for 35 years, and from -2.85 to -0.29 days/year for 40 years. The mean values of Theil-Sen (TS) slopes are -1.66, -1.41, and -1.33 for datasets covering the periods 1992–2020, 1987–2020, and 1982–2020, respectively.

The maps provided in Fig 6 display the spatial distribution of Theil-Sen slope values for the 30-year series (Fig 6a), 35-year series (Fig 6b), and 40-year series (Fig 6c), accompanied by histograms illustrating the Theil-Sen slope values. In the southern region of Poland, moderate downward trends were observed in the eastern part (represented by blue and green triangles), while higher values were evident in the central region (denoted by yellow and red triangles)

**Table 2. The results of the trend analysis; $Z_{MK}$ is the test statistic of the Mann-Kendall test, asterisk ↓ shows a downward trend, and · —No trend.**

| No | River/Gauging cross-section | 30 hydrological years, 1992–2021 | | | 35 hydrological years, 1987–2021 | | | 40 hydrological years, 1982–2021 | | |
|---|---|---|---|---|---|---|---|---|---|---|
| | | $Z_{MK}$ | Theil-Sen slope [day/year] | trend | $Z_{MK}$ | Theil-Sen slope [day/year] | trend | $Z_{MK}$ | Theil-Sen slope [day/year] | trend |
| 1 | Biała/Ciężkowice | -2.39 | -1.24 | ↓ | | | | | | |
| 2 | Biała/Grybów | -3.77 | -2.75 | ↓ | -2.03 | -0.78 | ↓ | -2.40 | -0.80 | ↓ |
| 3 | Biały Dunajec/Szaflary | -3.99 | -2.00 | ↓ | -4.35 | -2.44 | ↓ | -4.39 | -2.04 | ↓ |
| 4 | Czarna/Polana | -2.09 | -1.29 | ↓ | -4.35 | -1.95 | ↓ | -5.25 | -2.25 | ↓ |
| 5 | Czarna Orawa/Jabłonka | -3.12 | -1.67 | ↓ | -2.20 | -0.97 | ↓ | -2.67 | -1.15 | ↓ |
| 6 | Dunajec/Nowy Targ-Kowaniec | -1.68 | -1.00 | · | -3.30 | -1.33 | ↓ | -3.75 | -1.37 | ↓ |
| 7 | Jasiołka/Zboiska | -1.64 | -1.13 | · | -1.63 | -0.75 | · | -2.40 | -0.91 | ↓ |
| 8 | Kamienica/Łabowa | -1.25 | -0.90 | · | -1.78 | -1.00 | · | -1.83 | -0.82 | ↓ |
| 9 | Kamienica/Nowy Sącz | -2.07 | -1.00 | ↓ | -1.63 | -0.75 | · | -1.66 | -0.70 | · |
| 10 | Lepietnica/Ludźmierz | -1.46 | -0.87 | · | -1.96 | -0.80 | ↓ | -2.59 | -0.88 | ↓ |
| 11 | Łososina/Jakubkowice | -1.87 | -1.00 | ↓ | -1.19 | -0.60 | · | -1.48 | -0.60 | · |
| 12 | Łubinka/Nowy Sącz | -1.93 | -1.25 | ↓ | -1.39 | -0.70 | · | -1.75 | -0.73 | ↓ |
| 13 | Niedziczanka/Niedzica | -4.55 | -3.00 | ↓ | -2.08 | -1.00 | ↓ | -2.86 | -1.16 | ↓ |
| 14 | Olza/Istebna | -4.47 | -3.13 | ↓ | -4.67 | -2.50 | ↓ | -5.01 | -2.50 | ↓ |
| 15 | Osława/Szczawne | -1.68 | -0.95 | · | -4.64 | -2.78 | ↓ | -5.05 | -2.76 | ↓ |
| 16 | Pielnica/Nowosielce | -2.05 | -1.29 | ↓ | -1.91 | -0.81 | ↓ | -2.53 | -0.87 | ↓ |
| 17 | Poprad/Muszyna | -1.89 | -1.15 | ↓ | -1.98 | -1.00 | ↓ | -2.75 | -1.10 | ↓ |
| 18 | Poprad/Muszyna-Milik | -2.07 | -1.14 | ↓ | -2.30 | -1.08 | ↓ | -2.88 | -1.16 | ↓ |
| 19 | Poprad/Stary Sącz | -2.23 | -1.24 | ↓ | -2.34 | -1.00 | ↓ | -3.04 | -1.14 | ↓ |
| 20 | Raba/Kasinka Mała | -2.12 | -1.33 | ↓ | -2.51 | -1.13 | ↓ | -3.19 | -1.23 | ↓ |
| 21 | Raba/Rabka 2 | -3.54 | -2.63 | ↓ | -2.15 | -1.04 | ↓ | | | |
| 22 | Raba/Stróża | -3.50 | -2.31 | ↓ | -3.44 | -2.00 | ↓ | -2.93 | -1.20 | ↓ |
| 23 | Ropa/Klęczany | -1.21 | -0.71 | · | -3.37 | -1.89 | ↓ | -3.91 | -1.95 | ↓ |
| 24 | San/Przemyśl | -2.27 | -1.11 | ↓ | -0.82 | -0.38 | · | -3.16 | -1.50 | ↓ |
| 25 | San/Zatwarnica | -1.89 | -1.00 | ↓ | -2.13 | -0.80 | ↓ | -1.19 | -0.46 | · |
| 26 | Skawa/Sucha Beskidzka | -1.54 | -0.71 | · | -0.85 | -0.29 | · | -2.62 | -0.89 | ↓ |
| 27 | Solinka/Terka | -1.16 | -0.71 | · | -0.94 | -0.47 | · | -0.91 | -0.29 | · |
| 28 | Soła/Cięcina | -2.07 | -1.13 | ↓ | -2.16 | -0.86 | ↓ | -0.93 | -0.47 | · |
| 29 | Soła/Rajcza | -2.27 | -1.20 | ↓ | -1.88 | -0.84 | ↓ | -2.42 | -0.81 | ↓ |
| 30 | Strwiąż/Krościenko | -3.07 | -2.29 | ↓ | -2.90 | -1.54 | ↓ | -2.21 | -0.84 | ↓ |
| 31 | Stryszawka/Sucha Beskidzka | -0.64 | -0.56 | · | -0.97 | -0.50 | · | -3.07 | -1.26 | ↓ |
| 32 | Tanew/Harasiuki | -2.82 | -1.61 | ↓ | -2.96 | -1.38 | ↓ | -1.68 | -0.69 | · |
| 33 | Wetlina/Kalnica | -2.23 | -1.31 | ↓ | -2.39 | -1.13 | ↓ | -3.44 | -1.38 | ↓ |
| 34 | Wielki Rogoźnik/Ludźmierz | -4.89 | -3.72 | ↓ | -4.88 | -3.22 | ↓ | -3.08 | -1.20 | ↓ |
| 35 | Wilczka/Wilkanów | -2.05 | -0.80 | ↓ | -1.41 | -0.57 | · | -4.91 | -2.85 | ↓ |
| 36 | Wisłok/Puławy | -1.75 | -1.00 | · | -1.51 | -0.62 | · | | | · |
| 37 | Wisłok/Żarnowa | -1.32 | -0.77 | · | -1.71 | -0.81 | · | -1.04 | -0.32 | · |
| 38 | Wisłoka/Krajowice | -3.36 | -2.12 | ↓ | -2.87 | -1.43 | ↓ | -2.13 | -0.86 | ↓ |
| 39 | Wołosaty/Stuposiany | -2.07 | -1.20 | ↓ | -2.43 | -1.05 | ↓ | -2.56 | -1.00 | ↓ |
| 40 | Żabniczanka/Żabnica | -1.95 | -1.08 | ↓ | -1.79 | -0.78 | · | -3.09 | -1.32 | ↓ |

during the 30-year analysis. However, when extending the analyzed period, there is greater variability in the Theil-Sen slope values across different cross-sections. This variation highlights that the most significant trend in ice reduction has occurred in the central part of southern Poland over the past 30 years.

**Table 3. The number of gauging cross-sections where a statistically significant decreasing trend was identified.**
The percentage refers to the datasets in Table 2.

| Selected datasets / Trend | Stations with decreasing trends (no positive trends were detected) |
|---|---|
| 30 hydrological years | 29 |
| | 72,5% |
| 35 hydrological years | 26 |
| | 66,7% |
| 40 hydrological years | 30 |
| | 81,1% |
| 50 hydrological years | 1 |
| | 100% |

The statistically significant decreasing trends in the annual number of ice days are likely attributed to the rising air temperatures during the study period. An increasing trend in mean annual air temperature (observed across all stations) and mean winter air temperature (observed in most stations) was identified at a significance level of $\alpha = 0.05$. In contrast, no significant trends were observed in annual precipitation totals or maximum daily precipitation during the analyzed year/winter seasons, suggesting that changes in ice days may not be influenced by precipitation patterns.

Furthermore, an examination of water temperature changes using data from three available gauging stations (Raba in Stróża, Dunajec in Nowy Targ Kowaniec, and Biała in Grybów) revealed an increasing trend in mean annual water temperature (Raba/Stróża, Biała/Grybów) and mean winter water temperature (Biała/Grybów). This points towards the decreasing ice days in southern Poland being a consequence of rising air and water temperatures, aligning with regional climate change analyses in Poland that recognize the Carpathians as an area experiencing rapid warming [49].

The Mann-Kendall test results for series exceeding 50 years exhibit decreasing trends across all cross-sections. The Theil-Sen slopes range from -1.23 [days/year] in Ropa/Topoliny to

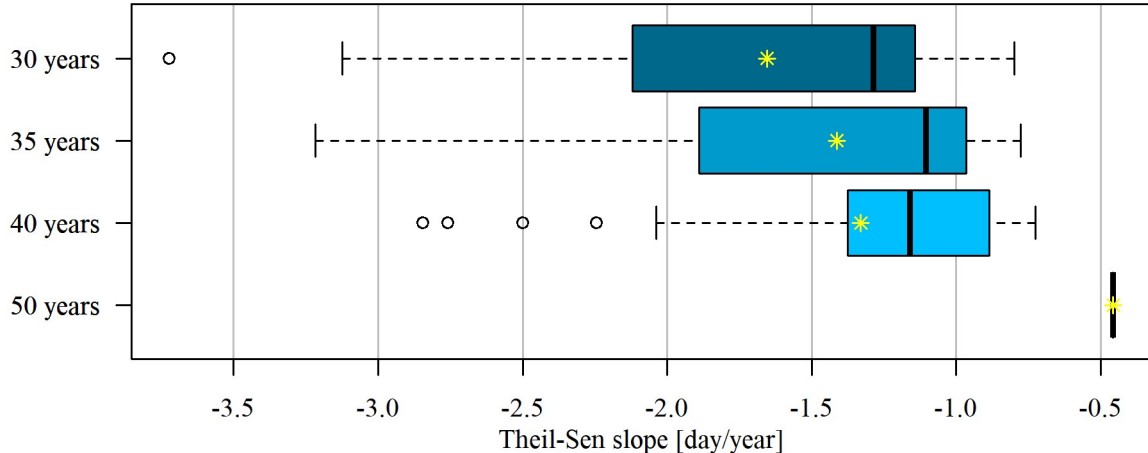

**Fig 5. The Theil-Sen slope estimator for the series of length (a) 30 years, (b) 35 years, and (c) 40 years where the trend was detected.** The vertical lines (in the box) denote from the left: Quantile 25%, median, and quantile 75%; the asterisks represent the mean values.

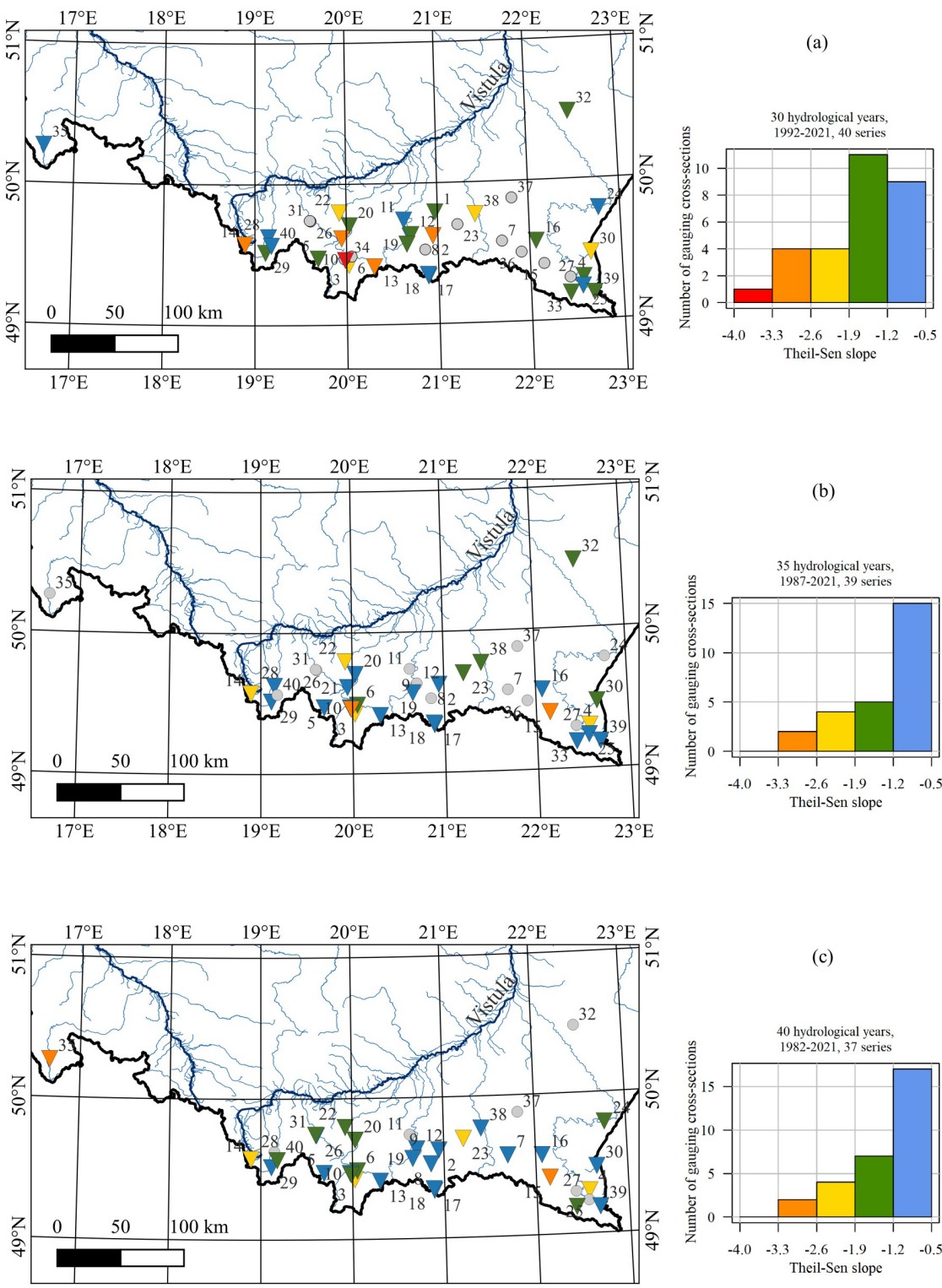

**Fig 6. The values of the Theil-Sen estimator for the series of lengths (a) 30 years, (b) 35 years, and (c) 40 years.** The triangles' sharp points indicate the downward trends while grey circles show cross sections with no trend. The strengths of the trend (Theil-Sen slope values) are reflected in the triangles' colors and can be identified in the histograms on the right side. Grey points show the location of the synoptic stations.

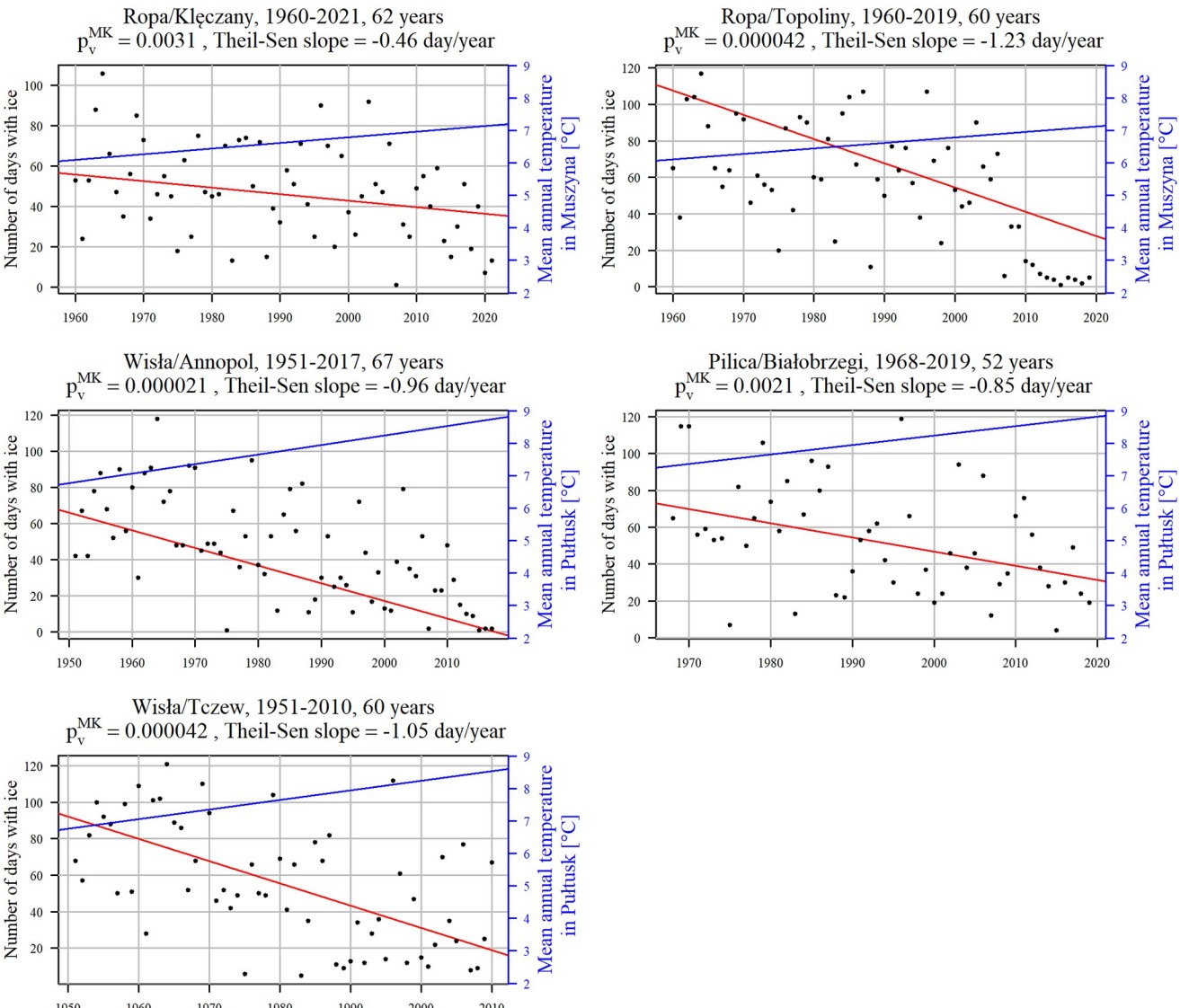

**Fig 7. The time series plots of the number of days with ice (red lines) and mean annual temperature (blue lines) for the series longer than 50 years.** The *p*-values of the MK test, the index number of days with ice, are denoted by $p_v^{MK}$. The red regression lines have slope values equal to the Theil-Sen slopes.

-0.46 [days/year] in Ropa/Klęczany (refer to Fig 7). Boxplots depicting the distribution of ice days concerning catchment areas (S1 Fig) and station altitudes (S2 Fig) were presented. Notably, there is no apparent pattern indicating a relationship between distribution and catchment area or station altitude observed in the plots. Additionally, the Theil-Sen slope value (refer to Table 2) of the trend appears to be unrelated to catchment area or station altitude, suggesting that orography is not a key determinant in the reduction of river ice duration and that the consequences of climate change on upland and mountainous catchments appear to be similar.

## Correlation and regression analysis

The results indicate that the Pearson correlation coefficient between the number of days with ice and the mean winter temperature is negative, ranging from -13.4 to -4.5. These two

variables exhibit a strong negative correlation across all stations (1992–2021), with all *p*-values < 0.05. This provides empirical support that the increase in mean winter temperature was the primary factor contributing to the decrease in ice days.

The outcomes of the linear regression analysis, utilizing the model $X = aT + b + \varepsilon$ (where *X* represents the number of days with ice, *T* denotes the mean winter temperature, and $\varepsilon$ signifies residuals), reveal that both *a* and *b* significantly differ from zero. The value of *a* varies from -22.04 to -13.57 during the period 1992–2021. The series of residuals $\varepsilon$ underwent testing for normality of distribution, with results from the Shapiro-Wilk test indicating normal distribution assumption holds for all but three stations. By considering the rate of change reflected by $\alpha$, one can infer that a 1˚C rise in mean winter temperature led to a reduction of ice days by 12 to 20 days.

Examining the spatial distribution of the two characteristics, it was observed that the most pronounced relationship between the NDI and temperature for the period 1992–2021 was evident in the southern part of the study area, with a somewhat weaker correlation observed in the southeast. However, it is prudent to interpret this observation cautiously as only data from nine synoptic stations were available for this period.

## Discussion

The analysis indicated a significant reduction in the annual number of days with ice phenomena in southern Polish rivers since the commencement of national hydrological monitoring. The most substantial change in Polish rivers was noted in the recent periods analyzed, highlighting a marked response of this phenomenon to climate change. While one might expect more pronounced differences in the annual ice days in datasets spanning longer periods, particularly during periods with significant disparities in air temperature indicators, our findings reveal that the last 30 years witnessed the most dynamic and abrupt increases in air temperatures, leading to a decrease in icing periods in rivers.

Assuming that the average annual sum of days with ice in Polish rivers during the analyzed periods is 55 days and extrapolate the results with the assumption that historical trends will continue at the same magnitudes in the future (ranging from -3.72 to -0.56 days/year in the 30-year analysis, -3.22 to -0.29 days/year in the 35-year analysis, and -2.85 to -0.29 days/year in the 40-year analysis), we can speculate that under the most conservative scenario, ice phenomena in Polish rivers may vanish in a few decades. This estimation considers a steady decline in ice days toward their likely disappearance by the mid-21st century based on the constant Theil-Sen slope.

When examining the potential impact of dam reservoirs on downstream sections with regards to ice formation [20,50], it was found that the locations of gauging cross-sections downstream of dam reservoirs are considerably distant from the dams. For instance, Klęczany at the River Ropa is approximately 26 km from the Klimkówka Dam, Topoliny at the River Ropa is around 52 km from the Klimkówka Dam, and Przemyśl at the River San is over 146 km from the Myczkowce Dam. The operational effect of dams on ice duration is deemed insignificant, attributing the decreasing trends in ice days primarily to climate change. The Klimkówka dam, put into operation in 1994, is located 54.4 km upstream from the river confluence. The distances between the two cross sections analyzed in this paper, namely Klęczany and Topoliny, and the river confluence are 26.9 km and 3 km, respectively. The impact of the Klimówka dam on ice occurrence in the River Ropa in the Klęczany and Topoliny cross-sections, was analysed by means of the MK test for the number of days with ice before and after the dam was constructed in 1994, i.e. 1960–1993, 1994–2021 for Klęczany and 1960–1993, 1994–2019 for Topoliny. The TS slope values in the two periods were -0.25 and

-1.11 (Klęczany) and -0.38 and -2.91 (Topoliny), respectively. The hypothesis about higher decrease in number of days with ice (i.e. lower TS slope) in the second period then in the first one was verified with bootstrap method. The results for Klęczany showed no trend in 1961–1993 and a decreasing trend in 1994–2021, however, the difference between the TS slope values is not significant suggesting non-substantial impact of the dam construction on the number of days with ice. Similarly, in Topoliny no trend was detected in 1961–2021 and a decreasing one in 1994–2019. This time, however, the difference between the TS slope values is significant being stronger after the construction of the dam. A deeper insight into the sample values showed the decreasing trend in the second period stemming from several anomalously low values in the years 2010–2019 (see Fig 7 for the Ropa/Topoliny cross-section that direct down the regression line) which also cause a strong decrease in the TS slope.

Future climate change impacts projected to affect water resources in Polish catchments, such as decreased snow accumulation and a decline in baseflow [15,51], do not appear to align with the observed general decrease in ice days in southern Polish rivers. This decline occurs uniformly across the analyzed region with consistent, though varying, intensities, indicating that the increase in winter temperatures is anticipated to have a more pronounced and unilateral effect compared to other seasons. Consequently, the warming of winters is expected to drive a widespread reduction in icing durations, likely evolving into a nationwide phenomenon independent of catchment physiography or human intervention levels.

From a water management perspective, the outcomes of our study offer insights into planning strategies to mitigate ice-related issues in rivers and address technological challenges associated with managing icing occurrences, such as protecting bridges and dam structures. The diminishing ice phenomena is seen as economically favorable in reducing the risk of water structure damage caused by ice flow. However, it is essential to recognize that adapting to the transition as ice events become less frequent yet still possible in colder winters is crucial to prevent potential damages incurred by river icing.

In terms of aquatic ecosystem protection, the declining trends in ice phenomena are anticipated to have adverse long-term implications, particularly for ecosystems dependent on ice processes. The alteration in ice patterns may disrupt crucial ecological processes, such as the formation of anastomosing riverbeds and the cleaning of river bottoms, potentially leading to habitat changes and impacting the breeding success of fish species, e.g. in the River Narew [52]. As the disappearing ice alters habitats and species dynamics, there may be cascading effects on aquatic plant species, life cycles of aquatic organisms, and biogeochemical processes in watercourses [53–58]. Consequently, the absence of ice phenomena could influence nutrient dynamics, potentially accelerating nutrient runoff and eutrophication in downstream water bodies. These comprehensive changes are designed to enhance the clarity, coherence, and accuracy of the original text while effectively conveying the given information in a more polished manner.

## Conclusions

The analysis conducted in this study has documented a noteworthy reduction in the duration of ice events in southern Poland. The observed shortening of the icing period in the examined rivers, contingent upon the length of datasets analyzed, spanned from -3.79 days/year to -0.29 days/year. Both mountain catchments (characterized by dynamic flow conditions) and lowland catchments (exhibiting slower flow conditions) are similarly affected by these diminishing icing periods. By examining air temperatures and exploring correlations between trends in air and water temperatures, we concluded that the decline in ice events on rivers in southern Poland is primarily a result of ongoing climate warming and not directly linked to human

activities. Notably, a nonlinear trend in this process has been identified, with a more rapid decrease in ice duration observed in recent years (1992–2021) compared to longer time frames. A 1˚C rise in mean winter temperature led to a reduction of 12–20 days with ice phenomena in the study area during the period 1992–2021. Climate variables such as precipitation did not display a significant impact on the occurrence of ice events. Should the observed trends persist, it is hypothesized that rivers in southern Poland will likely cease to exhibit ice phenomena around 2040–2070.

The gradual disappearance of ice in rivers undoubtedly disrupts the regularity of the hydrological cycle, causing changes in water accumulation and altering conditions for aquatic flora and fauna development in our climatic region. Given the escalated disappearance of ice events, particularly in the last 30 years, in Poland and potentially Central Europe, the comprehensive impacts remain largely unknown. Recognizing the complexity of these consequences, it is vital to concentrate on analyzing how individual components of aquatic ecosystems respond to the absence of ice phenomena. The intricate implications of this disappearance on rivers in Poland and Central Europe as a whole have yet to be fully understood. Looking ahead, conducting studies on the onset and cessation of ice events could provide valuable insights into changes in ice occurrence duration. Furthermore, exploring novel statistical methods for trend detection could be beneficial, although their efficacy should be rigorously evaluated before implementation.

## Supporting information

**S1 Fig. Distributions of the number of days with ice versus the catchment area enclosed by the gauging station.**
(TIF)

**S2 Fig. Distributions of the number of days with ice versus the altitude of the gauging station.**
(TIF)

**S1 Data. Datasets used in the research—Compressed files archive.**
(ZIP)

**S1 Appendix.**
(DOCX)

## Acknowledgments

The data available in the IMGW-PIB repositories enable further and deeper analysis of ice phenomena in Poland. Dr Barbara Nowicka is acknowledged for her help in data validation and acquisition. This work was supported by a subsidy from the Polish Ministry of Science and Higher Education for the Warsaw University of Technology, University of Agriculture in Cracow, Cracow University of Technology, Institute of Geophysics, Polish Academy of Sciences and University of Life Sciences in Warsaw (SGGW).

## Author Contributions

**Conceptualization:** Krzysztof Kochanek, Dorota Mirosław-Świątek, Mateusz Grygoruk.

**Data curation:** Krzysztof Kochanek, Agnieszka Rutkowska, Katarzyna Baran-Gurgul.

**Formal analysis:** Krzysztof Kochanek, Agnieszka Rutkowska, Katarzyna Baran-Gurgul, Iwona Kuptel-Markiewicz, Dorota Mirosław-Świątek.

**Funding acquisition:** Krzysztof Kochanek.

**Investigation:** Krzysztof Kochanek, Agnieszka Rutkowska, Katarzyna Baran-Gurgul, Iwona Kuptel-Markiewicz, Mateusz Grygoruk.

**Methodology:** Krzysztof Kochanek, Agnieszka Rutkowska, Katarzyna Baran-Gurgul, Iwona Kuptel-Markiewicz, Mateusz Grygoruk.

**Project administration:** Krzysztof Kochanek.

**Supervision:** Krzysztof Kochanek.

**Validation:** Krzysztof Kochanek, Agnieszka Rutkowska, Katarzyna Baran-Gurgul, Dorota Mirosław-Świątek, Mateusz Grygoruk.

**Writing – original draft:** Krzysztof Kochanek, Agnieszka Rutkowska, Katarzyna Baran-Gurgul, Iwona Kuptel-Markiewicz, Dorota Mirosław-Świątek, Mateusz Grygoruk.

**Writing – review & editing:** Krzysztof Kochanek, Agnieszka Rutkowska, Katarzyna Baran-Gurgul, Iwona Kuptel-Markiewicz, Mateusz Grygoruk.

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
