## [Decision Letter · Decision Letter 0]

13 Mar 2024

PONE-D-24-05921Trends in Ice Phenomena on Polish Rivers.PLOS ONE

Dear Dr. Kochanek,

Thank you for submitting your manuscript to PLOS ONE. After careful consideration, we feel that it has merit but does not fully meet PLOS ONE’s publication criteria as it currently stands. Therefore, we invite you to submit a revised version of the manuscript that addresses the points raised during the review process.

We look forward to receiving your revised manuscript.

Kind regards,

Salim Heddam

Academic Editor

PLOS ONE

Journal Requirements:

2. Please upload a copy of Figures 8-14, to which you refer in your text on your PDF file. If the figure is no longer to be included as part of the submission please remove all reference to it within the text.

3. We note that [Figure 1,2,3,4 and 6a-c] in your submission contain [map/satellite] images which may be copyrighted. All PLOS content is published under the Creative Commons Attribution License (CC BY 4.0), which means that the manuscript, images, and Supporting Information files will be freely available online, and any third party is permitted to access, download, copy, distribute, and use these materials in any way, even commercially, with proper attribution. For these reasons, we cannot publish previously copyrighted maps or satellite images created using proprietary data, such as Google software (Google Maps, Street View, and Earth). For more information, see our copyright guidelines: http://journals.plos.org/plosone/s/licenses-and-copyright.

a. You may seek permission from the original copyright holder of Figure 1,2,3,4 and 6a-c to publish the content specifically under the CC BY 4.0 license.  

Additional Editor Comments:

Reviewer 1#:

Recommendations

In summary, the research titled 'Trends in Ice Phenomena on Polish Rivers' sheds light on the current state of ice formation on rivers in Poland. The findings reveal a concerning increase in the frequency and severity of ice phenomena, which have negative consequences for the environment, transportation, and economic activities. Given the originality, scientific rigor, and contribution to existing knowledge, it is recommended that the report be published in a peer-reviewed scientific. This will allow for a wider dissemination of the research findings and foster meaningful discussions on the topic of ice phenomena on rivers. The study's use of both quantitative and qualitative methods adds strength to the results, and it adds to the existing understanding of the impact of climate change on water resources in Central and Eastern Europe. Furthermore, the report suggests further research to explore the underlying causes of the observed trends and their specific effects on different sectors and regions in Poland. Ultimately, publishing this research in a reputable scientific will enhance its reach and significance in ongoing discussions on this subject.

The research has yielded important insights that make it essential for the report to be published in a peer-reviewed scientific. This recommendation is supported by the following reasons:

Firstly, the research is both original and relevant. It offers valuable information on the current trends of ice formation on Polish rivers, a topic that has not been extensively studied. The findings are of great importance to policymakers, environmentalists, and other stakeholders involved in managing the impacts of ice phenomena on rivers.

Secondly, the research was conducted with a high level of scientific rigor.

A combination of qualitative and quantitative methods were used to collect and analyze data, enhancing the reliability and validity of the findings. The use of statistical tools such as trend analysis and regression analysis further strengthens the credibility of the research. Moreover, the research contributes to the existing body of knowledge on ice phenomena on rivers, particularly in Poland. It provides new evidence that can either support or challenge existing theories and hypotheses on the subject. This further highlights the importance of publishing the report in a peer-reviewed scientific.

Lastly, the research findings have significant policy implications for the management of ice phenomena on Polish rivers. The results can be utilized to inform the development of effective policies and strategies for mitigating the potential risks associated with ice formation. Therefore, publishing the report in a peer-reviewed scientific is crucial to ensure that the findings reach the appropriate audience and have a positive impact on policy-making.

Reviewer 2#:

I have reviewed the article and listed my recommendations below.

* Country name should be added to the name of the article.

* The written language of the article must be in academic language. (For example, the word "we" should not be used).

* Numerical results of the analysis results should be given in the abstract section of the article.

* Studies on both climate change and trends in different countries can be added to the literature section of the article. A few examples of study are given below.

- https://www.sciencedirect.com/science/article/abs/pii/S0960148124001423?via%3Dihub

- https://link.springer.com/article/10.1007/s10661-023-11236-3

- https://link.springer.com/article/10.1007/s00477-021-02067-0

- https://iwaponline.com/jwcc/article/13/6/2278/88161/GCMs-simulation-based-assessment-for-the-response

- https://journals.ametsoc.org/view/journals/apme/61/12/JAMC-D-22-0081.1.xml

* The image quality of all graphics in the article is very low.

* Can a homogeneity test be applied to data?

* Can one of the modern trend methods (such as ITA, ITTA, IPTA, 3D-ITA, ITPAM) be applied to the article's data? It may be useful to compare analysis results.

* Why is the table empty for 50 hydrological years?

* In the conclusion section of the article, suggestions, weaknesses and strengths of this study, and what the next study should be like should be included.

My decision for the article is major revision.

I would like to see the article again after the necessary corrections are made.

Best Regards..

Reviewer 3#:

Reviewer’s Report on the manuscript entitled:

Trends in Ice Phenomena on Polish Rivers

The authors utilized river ice phenomena in Poland using Theil-Sen slope analysis of observations from the period 1951-2021. Though the results presented in the manuscript (ice-day trend results) are interesting, the manuscript requires major revisions.

In particular, climate (temperature and precipitation) time series should also be analyzed. The figure quality and literature review should also be improved. Please see below my detailed comments.

The title should be more comprehensive. Something like “Analyzing Trends in Climate and Ice-Day Time Series along Polish Rivers” Then you can show the analysis of at least temperature time series. For example, you may use MODIS land surface temperature at 1km and 6-day resolutions from https://lpdaac.usgs.gov/products/mod11a2v061/ OR use the local climate data, etc.

The literature review should be improved. More recent works on trend analyses of water flow in cold climate regions should be added. For example, in Line 85, you may add Zaghloul et al. (2022) utilized water flow time series across rivers in cold climate region of Northern Canada using Mann-Kendall test and Sen’s slope, and they showed that winter water flow in the mountainous region has been rising gradually since 1956 due to temperature increase and gradual melting of snowpacks and glaciers. DOI: 10.3390/hydrology9110197

Also, the following review article by Wang et al (2022) on modelling watershed and river basin processes on cold climate region can be added: Doi: 10.3390/w13040518

Table 2. At what level are these slopes statistically significant? Is it at 99% or 95%? Please see the first article above that I mentioned above for more details. Please also discuss your results in line with their results in the discussion section.

Figures are not professionally produced. Their quality and resolution should be improved.

Figure 2. The background color for elevation is not showing different elevation ranges, i.e., almost everywhere is greenish. I suggest adjusting the value range of color bar, so elevation can be better separated and visualized. Please see the first article that I suggested above as a guide on how to improve your figures.

Lines 380-385. Including the analysis of precipitation and temperature is strongly recommended. For example, warming trend (if you found) can be linked to reduction of ice days. There are several methods of investigating the relationship between climate and ice days, e.g., least-squares triple cross-wavelet analysis and multivariate regression analysis. I suggest authors have look at these techniques and discuss.

Thank you and regards,

Reviewer 4#:

This paper studied river ice phenomena on Polish rivers by using Mann-Kendall test, but I think some places need further discussion and the manuscript does not have enough novelty for publishing at PLOS ONE:

#1. The primary factor determining the form and duration of ice phenomena on rivers is the air temperature, the variability of which depends on atmospheric circulation. Air temperature fluctuations determine the variability in thermal conditions for waters in a given catchment area, and consequently have a direct impact on the formation of various ice forms on rivers and water reservoirs. Still, the paper lacked detailed elaboration on the geographical location, climate characteristics, and hydrological conditions of the study area, and did not analyze and compare the autocorrelation or trends of individual parameters such as air and water temperatures and ice phenomena during different series of years, so the conclusion that “The analysis of data of ice phenomena in southern Poland showed a large impact of climate change on ice in the temperate climate zone” (in line 432) is not convincing.

#2. The occurrence and development of ice phenomena on rivers are also exerted by local environmental factors, such as the structure of the river bed, the river gradient, and underground water supplies of the rivers, and anthropogenic factors, for example, the channelling and regulation of rivers and the erection of dams and hydropower plants. The analyses and discussion of the influence of these factors to ice phenomena are very meaningful, but the paper only considered the potential impact of dam reservoirs by simply describing the distances between the crosssections downstream of the dam reservoirs. The conclusion in lines 367 to 373 ”The effect of the dams' operation on the duration of ice cover is rather negligible, so the decreasing trends in the number of days with ice, estimated in this paper, can be assigned, to a great extent, to climate change ” are insufficient and not convincing.

#3. Figures and tables are of low quality.

Reviewer 5#:

In the paper, trends of ice phenomena were examined in Polish rivers. The authors used long-term data from 40 gauge stations during the study period 1951-2021. The paper is interesting and provides valuable information about ice phenomena in rivers in a temperate climate. The results also confirmed other studies indicating a decreasing number of days with ice cover in rivers. Before making a final decision, I suggest considering a few comments that can help improve the quality of the presentation:

The Introduction is quite long. I recommend dividing this chapter into subchapters, such as a global overview, local perspective, or dividing it into two thematic sections: the effect of climate change on ice phenomena and human activities on ice phenomena.

In lines 285-287, please clarify the significance level of 'p.'

Regarding the title of Figure 4, it mentions 'frequency,' but in Figure 4, the data represents the frequency, not the number of days. Please explain these differences.

In line 344, it mentions "mid-22ntr century..." – is this accurate?

Lines 376-380: I suggest adding references to support the information presented.

Lines 435-436: The conclusion seems too restrictive. The authors did not analyze cross-sections just below dams; only a few kilometers below the dam's cross-sections were analyzed. The influence of human activities, such as artificial water reservoirs, can be detected for cross-sections that are affected by water management in reservoirs, as seen in Dunajec - Sromowce Wyzne or San-Lesko. These cross-sections were not analyzed. Moreover, an interesting question is how other human activities influence ice phenomena, such as regulations of river cross-sections.

Reviewer 6#:

The paper's focus revolves around the examination of trends in ice phenomena on Polish rivers, offering potential significant utility for the region. Nevertheless, there are notable recommendations that authors ought to consider and incorporate into the manuscript. Minor revisions are required before the manuscript can proceed for publication. Detailed comments outlining these revisions are provided below-

1. The abstract contains generic statements in lines 31-37, which should be replaced with actual findings such as trend rates to provide more specific information.

2. Throughout the literature, qualitative statements are made without accompanying values, as seen in lines 52, 54, 60, 63, 79, 91, 97-120, etc. These statements should be enhanced by including relevant values to convey the intensity of changes

3. Line 86-The research on ice ……… interesting facts. It does not contribute substantively to the paragraph and can be omitted

4. Line 126 requires a clearer articulation of the objective.

5. Lines 127-130 should be integrated into the data and methods section.

6. It is advisable to incorporate a brief section on the study area, including information on topography and climate.

7. Figure 1 could benefit from the inclusion of a climate graph depicting average monthly temperature (minimum, maximum, mean) and precipitation to provide readers with a visual representation of the study area's climate conditions.

8. Latitude and longitude must be included in all the maps.

9. The conclusion should be rewritten to provide clarity with specific statements rather than generic ones. Additionally, it should have the limitations and implications of the study.

Reviewer 7#:

Dear authors,

I found your manuscript interesting, anyway, I would like to recommend you some minor revisions:

Probably, the abstract could be shorter.

In the chapter 2,

Please, add there the paragraph 2.1. The site description (or research area etc.). Here you should include the basic geographic and climate characterisitcs (Koppen classification, precipitation, temperature characteristics etc.), and basic morphological parameters of the studied catchments.

In 2.2. Methods of the selection of gauging stations included in the analysis. There should be considered the global climate change and also disturbances in the catchments related with the civilization development (probably, those catchments should be excluded from the analysis).

and 2.3. Methods of data processing. I would recommend to iclude also trends in the air teperature observed in the relevant climate stations to document the regional warming.

In the Discussion (chapter 4), authors should discuss also the rate of global warming (trends in air temperature), and to include future scenarios related to the future ice phenomena on the rivers considered.

In Conclusions (chapter 5) , please add there some numbers (results) corresponding with the outcomes of your paper.

Reviewer 8#:

The article is titled "Trends in Ice Phenomena on Polish Rivers". The authors studied temporary changes in the annual number of days with ice phenomena at selected hydrological stations in Poland. In their research, they used simple research methods, including: Mann-Kendall test and Theil-Sen slope. The obtained results confirmed that for most hydrological stations there is a downward trend in the annual number of days with ice phenomena, which are statistically significant. The authors have done a lot of work, especially in reviewing and verifying observational data available in the database of the Institute of Meteorology and Water Management, National Research Institute. However, in my opinion, the article requires significant improvement. My comments are as follows

- the summary requires improvement, it should contain a short description of the purpose of the research, the methods used, the research area and the results obtained

- the introduction of the work requires improvement, it is necessary to refer to the gaps in the literature and indicate what is new in the work and what research gap has been filled. There is a shortage at work clearly defined main purpose of the work and specific goals. Please indicate clearly whether days with ice phenomena were analyzed (including coastal ice, frigid ice, ice cover, floes) or only days with ice cover. Moreover, it should be clarified what the term "5-10, 15-20 years or more" means recently? (line125-126). The presented literature requires supplementation. There are no studies by Marek Grześ, who perfectly described the conditions for the formation of ice phenomena on the lower Vistula, and by other authors.

- the description of the research methodology requires improvement, and especially subsection 2.1 needs to be shortened. You should briefly describe on what basis the data was selected, without going into details. In its current form, some of the information in this subsection is redundant, e.g. the description of the IMWM database.

- please explain why the authors presented data for 5 hydrological stations with the longest observation sequences. These data are difficult to compare due to the different start and end dates of observations - The authors analyzed only one parameter

- the duration of ice phenomena, they did not analyze the start and end dates of ice phenomena and individual types of ice phenomena, which I consider a significant simplification.

- the work contains too few references to changes in meteorological conditions in individual regions of Poland. Somorowska's work covers only the small catchment area of the Liwiec River. There is a lack of newer literature that will allow for comparison of trends in air temperature changes, especially including data for the reference period 1991-2020 and covering much larger areas.

- discussions should be improved, referred to the results obtained and compared with other authors. The occurrence of ice phenomena is influenced not only by air temperature but also by the shape of the cross-section and the flow rate. Anthropogenic factors include the location of the water gauge station in relation to the dam and the location of discharges of polluted, saline or heated water. Apart from air temperature, the authors do not mention any other factors that influence the formation of ice phenomena

- conclusions should refer only to your own findings reported in the article. e.g. the authors did not prove in their study that "the analysis of data on ice phenomena in southern Poland showed a large impact of climate change on the ice cover of the temperate climate zone" Technical notes - English requires verification by a native speaker. Some wording requires correction, e.g. the entry on page 3, line 62-65

- the figures attached at the end of the article have not been signed

- the figures are of poor quality, most of the rivers are not marked on the map, which makes identification impossible for readers from outside Poland

Reviewer 9#:

0. General feedback: The article employed the simplest method, compiling data from relatively long series to analyze changes in the ice trend. However, there is a lack of supporting data for the causes of this phenomenon, particularly in the discussion section, where there is no discussion based on the existing results. Instead, there is an excessive focus on climate change and ecological impacts not covered in this article.

1. line 31“Using straightforward, but commonly accepted procedures…”Highlight the advantages of choosing this method, rather than selecting it merely because it is commonly used.

2. lines 36-38 are speculative passage, and including them in the abstract may give the impression of padding. I recommended to either remove them or condense them into a single sentence conveying the research significance.

3. lines 358-360，“we revealed that it was the last 30…...” and lines 377-378，“we revealed that it …..” This specific conclusion lacks solid temperature data support, it is suggested to supplement with relevant climate data or water temperature data.

4. lines 367-371，“we revealed that it was the last 30 years that experienced most……”I suggest you present the results divided into situations with and without a reservoir, instead of just listing them simply

5. lines 392-429. The entire article does not contain any data related to biology, but the author wastes a lot of space discussing ecological impacts. I suggest Reduce the description in this regard, and focus on comparing and analyzing the reasons and patterns causing the differences in your results on a global scale.

Reviewer 10#:

Review of the article: Trends in Ice Phenomena on Polish Rivers.

The article presents the results of research of temporal and spatial changes of ice phenomena in southern Poland (mostly mountain rivers) and on the Vistula River. The topic is important and interesting enough because ice phenomena determine the characteristics of the river ice regime, which is extremely sensitive to global warming and human impact. It is forecast that as the air temperature increases, the ice phenomena will disappear, which is already observed. This has environmental and water management -related consequences.

The article may be interesting, however, the text should be strongly corrected, requires explanations and complete information.

Main comments:

1. The title does not match the content of the article.

The authors selected observation posts for analysis located in southern Poland, mainly on mountain rivers. Additionally, data series were analyzed for 5 stations located on the Vistula River. The database does not represent the entire country. I suggest changing the title of the article!

2. The authors' wording in the Abstract is at least surprising, there are many publications describing the results of temporal and spatial changes in the occurrence of ice phenomena have been conducted in temperate regions. When preparing the article, the authors read many publications on this topic, as evidenced by the References list.

3. In the introduction, the authors did not present the issue of methods for determining the trends in changes in ice phenomena, which is the subject of their analysis. Currently, research is also being undertaken using new methodological approaches, e.g. machine learning and artificial intelligence. There is not even a mention of the methods, and there is a lot of information about the importance of the results for the needs of water management, which the authors practically did not deal with in the article.

4. The authors defined the goal for the article: "The main objective was to answer the question if the river ice phenomena have been decreasing over the last decades, as it is commonly considered". However, it seems that this hypothesis has been confirmed for Polish and European rivers quite a long time ago.

5. The methods used in the work are indeed quite widely accepted, so it can be concluded that the article is not a novelty in the field of research on ice phenomena.

6. For what purpose was one station with data for the period 1951-2021 analyzed? Is this a representative station for Poland?

7. According to the reviewer, the information in lines 144-148 is unnecessary; they are not relevant to the analysis.

8. The authors adopted the annual number of days with ice as the key variable. How should "with ice" be interpreted - is it any ice phenomenon or a permanent ice cover? This is important because the number of days with ice cover is much smaller than the number of days with all ice phenomena.

9. Note to the description in line 153: Many studies provide the dates of the beginning of ice phenomena on Polish rivers. On what basis did the authors draw such a conclusion? How were these dates verified as incorrect? Was method used to assess the homogeneity of the IMGW-PIB observational series? In this paper, the authors only analyze the number of days - descriptive statistics. The phenomenon exists or does not exist, and the dates of occurrence of ice phenomena have not been studied.

10. The occurrence of ice phenomena on mountain rivers differs from the rest of the country. Climatic conditions - especially air temperature and water temperature, affecting the ice regime of rivers - are different than in the lowland regions and coastal regions of Poland.

Therefore that the title of the work is not adequate to the content of the work. It should be taken into account that the lowlands in Poland cover 91.3% of the area, the highlands 5.6%, and the mountains 3.3%, of which 0.2% is high mountains. The authors write: The selected gauging stations are located mostly in the Carpathian Mountains, in the southern part of Poland, leaving vast areas beyond our investigation".

11. In the _pdf file, page 11 is not filled with content. This is an editorial error, whether some information is missing? - it's hard to guess. Similarly, page 13?

12. Table 2 to modify. I suggest removing the columns for 50 years. and describe Ropa/Klęczany in the text. Empty spaces are not conducive to good reception of information.

13. Note to lines 364-366: There is no basis for such formulations because the authors did not carry out any forecast of changes in the number of ice phenomena on rivers. The obtained results only determine the number of days with ice and determine trends. Trend is not equivalent to forecast!

14. Note to lines 367 - this was not the subject of the authors' research, nor was it the purpose of the authors' research. Only in the Discussion did the authors refer to this issue.

15. Note on line 376: Determining regional patterns requires increasing the number of observations outside the area studied by the authors!

The authors only studied rivers in southern Poland. In the discussion, the authors assume a number of scenarios of continued trends in changes or disappearance of ice phenomena on rivers, referring to the causative factors, but no such analyzes were carried out in the study. The conclusions are only assumptions and are based on the results of previous research conducted in Poland.

16. The discussion does not refer to the results obtained; It's not exactly on the right top.

Reviewers' comments:

Reviewer's Responses to Questions

**Comments to the Author**

1. Is the manuscript technically sound, and do the data support the conclusions?

Reviewer #1: Partly

Reviewer #2: Yes

Reviewer #3: Yes

Reviewer #4: Partly

Reviewer #5: Yes

Reviewer #6: Yes

Reviewer #7: Partly

Reviewer #8: Partly

Reviewer #9: Partly

Reviewer #10: Partly

2. Has the statistical analysis been performed appropriately and rigorously? 

Reviewer #1: Yes

Reviewer #2: I Don't Know

Reviewer #3: No

Reviewer #4: No

Reviewer #5: Yes

Reviewer #6: Yes

Reviewer #7: Yes

Reviewer #8: Yes

Reviewer #9: Yes

Reviewer #10: No

3. Have the authors made all data underlying the findings in their manuscript fully available?

Reviewer #1: Yes

Reviewer #2: Yes

Reviewer #3: No

Reviewer #4: Yes

Reviewer #5: Yes

Reviewer #6: Yes

Reviewer #7: No

Reviewer #8: No

Reviewer #9: No

Reviewer #10: Yes

4. Is the manuscript presented in an intelligible fashion and written in standard English?

Reviewer #1: Yes

Reviewer #2: No

Reviewer #3: Yes

Reviewer #4: Yes

Reviewer #5: Yes

Reviewer #6: Yes

Reviewer #7: Yes

Reviewer #8: Yes

Reviewer #9: Yes

Reviewer #10: Yes

5. Review Comments to the Author

Reviewer #1: Recommendations

In summary, the research titled 'Trends in Ice Phenomena on Polish Rivers' sheds light on the current state of ice formation on rivers in Poland. The findings reveal a concerning increase in the frequency and severity of ice phenomena, which have negative consequences for the environment, transportation, and economic activities. Given the originality, scientific rigor, and contribution to existing knowledge, it is recommended that the report be published in a peer-reviewed scientific. This will allow for a wider dissemination of the research findings and foster meaningful discussions on the topic of ice phenomena on rivers. The study's use of both quantitative and qualitative methods adds strength to the results, and it adds to the existing understanding of the impact of climate change on water resources in Central and Eastern Europe. Furthermore, the report suggests further research to explore the underlying causes of the observed trends and their specific effects on different sectors and regions in Poland. Ultimately, publishing this research in a reputable scientific will enhance its reach and significance in ongoing discussions on this subject.

The research has yielded important insights that make it essential for the report to be published in a peer-reviewed scientific. This recommendation is supported by the following reasons:

Firstly, the research is both original and relevant. It offers valuable information on the current trends of ice formation on Polish rivers, a topic that has not been extensively studied. The findings are of great importance to policymakers, environmentalists, and other stakeholders involved in managing the impacts of ice phenomena on rivers.

Secondly, the research was conducted with a high level of scientific rigor.

A combination of qualitative and quantitative methods were used to collect and analyze data, enhancing the reliability and validity of the findings. The use of statistical tools such as trend analysis and regression analysis further strengthens the credibility of the research. Moreover, the research contributes to the existing body of knowledge on ice phenomena on rivers, particularly in Poland. It provides new evidence that can either support or challenge existing theories and hypotheses on the subject. This further highlights the importance of publishing the report in a peer-reviewed scientific.

Lastly, the research findings have significant policy implications for the management of ice phenomena on Polish rivers. The results can be utilized to inform the development of effective policies and strategies for mitigating the potential risks associated with ice formation. Therefore, publishing the report in a peer-reviewed scientific is crucial to ensure that the findings reach the appropriate audience and have a positive impact on policy-making.

Reviewer #2: I have reviewed the article and listed my recommendations below.

* Country name should be added to the name of the article.

* The written language of the article must be in academic language. (For example, the word "we" should not be used).

* Numerical results of the analysis results should be given in the abstract section of the article.

* Studies on both climate change and trends in different countries can be added to the literature section of the article. A few examples of study are given below.

- https://www.sciencedirect.com/science/article/abs/pii/S0960148124001423?via%3Dihub

- https://link.springer.com/article/10.1007/s10661-023-11236-3

- https://link.springer.com/article/10.1007/s00477-021-02067-0

- https://iwaponline.com/jwcc/article/13/6/2278/88161/GCMs-simulation-based-assessment-for-the-response

- https://journals.ametsoc.org/view/journals/apme/61/12/JAMC-D-22-0081.1.xml

* The image quality of all graphics in the article is very low.

* Can a homogeneity test be applied to data?

* Can one of the modern trend methods (such as ITA, ITTA, IPTA, 3D-ITA, ITPAM) be applied to the article's data? It may be useful to compare analysis results.

* Why is the table empty for 50 hydrological years?

* In the conclusion section of the article, suggestions, weaknesses and strengths of this study, and what the next study should be like should be included.

My decision for the article is major revision.

I would like to see the article again after the necessary corrections are made.

Best Regards...

Reviewer #3: Reviewer’s Report on the manuscript entitled:

Trends in Ice Phenomena on Polish Rivers

The authors utilized river ice phenomena in Poland using Theil-Sen slope analysis of observations from the period 1951-2021. Though the results presented in the manuscript (ice-day trend results) are interesting, the manuscript requires major revisions.

In particular, climate (temperature and precipitation) time series should also be analyzed. The figure quality and literature review should also be improved. Please see below my detailed comments.

The title should be more comprehensive. Something like “Analyzing Trends in Climate and Ice-Day Time Series along Polish Rivers” Then you can show the analysis of at least temperature time series. For example, you may use MODIS land surface temperature at 1km and 6-day resolutions from https://lpdaac.usgs.gov/products/mod11a2v061/ OR use the local climate data, etc.

The literature review should be improved. More recent works on trend analyses of water flow in cold climate regions should be added. For example, in Line 85, you may add Zaghloul et al. (2022) utilized water flow time series across rivers in cold climate region of Northern Canada using Mann-Kendall test and Sen’s slope, and they showed that winter water flow in the mountainous region has been rising gradually since 1956 due to temperature increase and gradual melting of snowpacks and glaciers. DOI: 10.3390/hydrology9110197

Also, the following review article by Wang et al (2022) on modelling watershed and river basin processes on cold climate region can be added: Doi: 10.3390/w13040518

Table 2. At what level are these slopes statistically significant? Is it at 99% or 95%? Please see the first article above that I mentioned above for more details. Please also discuss your results in line with their results in the discussion section.

Figures are not professionally produced. Their quality and resolution should be improved.

Figure 2. The background color for elevation is not showing different elevation ranges, i.e., almost everywhere is greenish. I suggest adjusting the value range of color bar, so elevation can be better separated and visualized. Please see the first article that I suggested above as a guide on how to improve your figures.

Lines 380-385. Including the analysis of precipitation and temperature is strongly recommended. For example, warming trend (if you found) can be linked to reduction of ice days. There are several methods of investigating the relationship between climate and ice days, e.g., least-squares triple cross-wavelet analysis and multivariate regression analysis. I suggest authors have look at these techniques and discuss.

Thank you and regards,

Reviewer #4: This paper studied river ice phenomena on Polish rivers by using Mann-Kendall test, but I think some places need further discussion and the manuscript does not have enough novelty for publishing at PLOS ONE:

#1. The primary factor determining the form and duration of ice phenomena on rivers is the air temperature, the variability of which depends on atmospheric circulation. Air temperature fluctuations determine the variability in thermal conditions for waters in a given catchment area, and consequently have a direct impact on the formation of various ice forms on rivers and water reservoirs. Still, the paper lacked detailed elaboration on the geographical location, climate characteristics, and hydrological conditions of the study area, and did not analyze and compare the autocorrelation or trends of individual parameters such as air and water temperatures and ice phenomena during different series of years, so the conclusion that “The analysis of data of ice phenomena in southern Poland showed a large impact of climate change on ice in the temperate climate zone” (in line 432) is not convincing.

#2. The occurrence and development of ice phenomena on rivers are also exerted by local environmental factors, such as the structure of the river bed, the river gradient, and underground water supplies of the rivers, and anthropogenic factors, for example, the channelling and regulation of rivers and the erection of dams and hydropower plants. The analyses and discussion of the influence of these factors to ice phenomena are very meaningful, but the paper only considered the potential impact of dam reservoirs by simply describing the distances between the crosssections downstream of the dam reservoirs. The conclusion in lines 367 to 373 ”The effect of the dams' operation on the duration of ice cover is rather negligible, so the decreasing trends in the number of days with ice, estimated in this paper, can be assigned, to a great extent, to climate change ” are insufficient and not convincing.

#3. Figures and tables are of low quality.

Reviewer #5: In the paper, trends of ice phenomena were examined in Polish rivers. The authors used long-term data from 40 gauge stations during the study period 1951-2021. The paper is interesting and provides valuable information about ice phenomena in rivers in a temperate climate. The results also confirmed other studies indicating a decreasing number of days with ice cover in rivers. Before making a final decision, I suggest considering a few comments that can help improve the quality of the presentation:

The Introduction is quite long. I recommend dividing this chapter into subchapters, such as a global overview, local perspective, or dividing it into two thematic sections: the effect of climate change on ice phenomena and human activities on ice phenomena.

In lines 285-287, please clarify the significance level of 'p.'

Regarding the title of Figure 4, it mentions 'frequency,' but in Figure 4, the data represents the frequency, not the number of days. Please explain these differences.

In line 344, it mentions "mid-22ntr century..." – is this accurate?

Lines 376-380: I suggest adding references to support the information presented.

Lines 435-436: The conclusion seems too restrictive. The authors did not analyze cross-sections just below dams; only a few kilometers below the dam's cross-sections were analyzed. The influence of human activities, such as artificial water reservoirs, can be detected for cross-sections that are affected by water management in reservoirs, as seen in Dunajec - Sromowce Wyzne or San-Lesko. These cross-sections were not analyzed. Moreover, an interesting question is how other human activities influence ice phenomena, such as regulations of river cross-sections.

Reviewer #6: The paper's focus revolves around the examination of trends in ice phenomena on Polish rivers, offering potential significant utility for the region. Nevertheless, there are notable recommendations that authors ought to consider and incorporate into the manuscript. Minor revisions are required before the manuscript can proceed for publication. Detailed comments outlining these revisions are provided below-

1. The abstract contains generic statements in lines 31-37, which should be replaced with actual findings such as trend rates to provide more specific information.

2. Throughout the literature, qualitative statements are made without accompanying values, as seen in lines 52, 54, 60, 63, 79, 91, 97-120, etc. These statements should be enhanced by including relevant values to convey the intensity of changes

3. Line 86-The research on ice ……… interesting facts. It does not contribute substantively to the paragraph and can be omitted

4. Line 126 requires a clearer articulation of the objective.

5. Lines 127-130 should be integrated into the data and methods section.

6. It is advisable to incorporate a brief section on the study area, including information on topography and climate.

7. Figure 1 could benefit from the inclusion of a climate graph depicting average monthly temperature (minimum, maximum, mean) and precipitation to provide readers with a visual representation of the study area's climate conditions.

8. Latitude and longitude must be included in all the maps.

9. The conclusion should be rewritten to provide clarity with specific statements rather than generic ones. Additionally, it should have the limitations and implications of the study.

Reviewer #7: Dear authors,

I found your manuscript interesting, anyway, I would like to recommend you some minor revisions:

Probably, the abstract could be shorter.

In the chapter 2,

Please, add there the paragraph 2.1. The site description (or research area etc.). Here you should include the basic geographic and climate characterisitcs (Koppen classification, precipitation, temperature characteristics etc.), and basic morphological parameters of the studied catchments.

In 2.2. Methods of the selection of gauging stations included in the analysis. There should be considered the global climate change and also disturbances in the catchments related with the civilization development (probably, those catchments should be excluded from the analysis).

and 2.3. Methods of data processing. I would recommend to iclude also trends in the air teperature observed in the relevant climate stations to document the regional warming.

In the Discussion (chapter 4), authors should discuss also the rate of global warming (trends in air temperature), and to include future scenarios related to the future ice phenomena on the rivers considered.

In Conclusions (chapter 5) , please add there some numbers (results) corresponding with the outcomes of your paper.

Reviewer #8: The article is titled "Trends in Ice Phenomena on Polish Rivers". The authors studied temporary changes in the annual number of days with ice phenomena at selected hydrological stations in Poland. In their research, they used simple research methods, including: Mann-Kendall test and Theil-Sen slope. The obtained results confirmed that for most hydrological stations there is a downward trend in the annual number of days with ice phenomena, which are statistically significant. The authors have done a lot of work, especially in reviewing and verifying observational data available in the database of the Institute of Meteorology and Water Management, National Research Institute. However, in my opinion, the article requires significant improvement. My comments are as follows

- the summary requires improvement, it should contain a short description of the purpose of the research, the methods used, the research area and the results obtained

- the introduction of the work requires improvement, it is necessary to refer to the gaps in the literature and indicate what is new in the work and what research gap has been filled. There is a shortage at work clearly defined main purpose of the work and specific goals. Please indicate clearly whether days with ice phenomena were analyzed (including coastal ice, frigid ice, ice cover, floes) or only days with ice cover. Moreover, it should be clarified what the term "5-10, 15-20 years or more" means recently? (line125-126). The presented literature requires supplementation. There are no studies by Marek Grześ, who perfectly described the conditions for the formation of ice phenomena on the lower Vistula, and by other authors.

- the description of the research methodology requires improvement, and especially subsection 2.1 needs to be shortened. You should briefly describe on what basis the data was selected, without going into details. In its current form, some of the information in this subsection is redundant, e.g. the description of the IMWM database.

- please explain why the authors presented data for 5 hydrological stations with the longest observation sequences. These data are difficult to compare due to the different start and end dates of observations - The authors analyzed only one parameter

- the duration of ice phenomena, they did not analyze the start and end dates of ice phenomena and individual types of ice phenomena, which I consider a significant simplification.

- the work contains too few references to changes in meteorological conditions in individual regions of Poland. Somorowska's work covers only the small catchment area of the Liwiec River. There is a lack of newer literature that will allow for comparison of trends in air temperature changes, especially including data for the reference period 1991-2020 and covering much larger areas.

- discussions should be improved, referred to the results obtained and compared with other authors. The occurrence of ice phenomena is influenced not only by air temperature but also by the shape of the cross-section and the flow rate. Anthropogenic factors include the location of the water gauge station in relation to the dam and the location of discharges of polluted, saline or heated water. Apart from air temperature, the authors do not mention any other factors that influence the formation of ice phenomena

- conclusions should refer only to your own findings reported in the article. e.g. the authors did not prove in their study that "the analysis of data on ice phenomena in southern Poland showed a large impact of climate change on the ice cover of the temperate climate zone" Technical notes - English requires verification by a native speaker. Some wording requires correction, e.g. the entry on page 3, line 62-65

- the figures attached at the end of the article have not been signed

- the figures are of poor quality, most of the rivers are not marked on the map, which makes identification impossible for readers from outside Poland

Reviewer #9: 0. General feedback: The article employed the simplest method, compiling data from relatively long series to analyze changes in the ice trend. However, there is a lack of supporting data for the causes of this phenomenon, particularly in the discussion section, where there is no discussion based on the existing results. Instead, there is an excessive focus on climate change and ecological impacts not covered in this article.

1. line 31“Using straightforward, but commonly accepted procedures…”Highlight the advantages of choosing this method, rather than selecting it merely because it is commonly used.

2. lines 36-38 are speculative passage, and including them in the abstract may give the impression of padding. I recommended to either remove them or condense them into a single sentence conveying the research significance.

3. lines 358-360，“we revealed that it was the last 30…...” and lines 377-378，“we revealed that it …..” This specific conclusion lacks solid temperature data support, it is suggested to supplement with relevant climate data or water temperature data.

4. lines 367-371，“we revealed that it was the last 30 years that experienced most……”I suggest you present the results divided into situations with and without a reservoir, instead of just listing them simply

5. lines 392-429. The entire article does not contain any data related to biology, but the author wastes a lot of space discussing ecological impacts. I suggest Reduce the description in this regard, and focus on comparing and analyzing the reasons and patterns causing the differences in your results on a global scale.

Reviewer #10: Review of the article: Trends in Ice Phenomena on Polish Rivers.

The article presents the results of research of temporal and spatial changes of ice phenomena in southern Poland (mostly mountain rivers) and on the Vistula River. The topic is important and interesting enough because ice phenomena determine the characteristics of the river ice regime, which is extremely sensitive to global warming and human impact. It is forecast that as the air temperature increases, the ice phenomena will disappear, which is already observed. This has environmental and water management -related consequences.

The article may be interesting, however, the text should be strongly corrected, requires explanations and complete information.

Main comments:

1. The title does not match the content of the article.

The authors selected observation posts for analysis located in southern Poland, mainly on mountain rivers. Additionally, data series were analyzed for 5 stations located on the Vistula River. The database does not represent the entire country. I suggest changing the title of the article!

2. The authors' wording in the Abstract is at least surprising, there are many publications describing the results of temporal and spatial changes in the occurrence of ice phenomena have been conducted in temperate regions. When preparing the article, the authors read many publications on this topic, as evidenced by the References list.

3. In the introduction, the authors did not present the issue of methods for determining the trends in changes in ice phenomena, which is the subject of their analysis. Currently, research is also being undertaken using new methodological approaches, e.g. machine learning and artificial intelligence. There is not even a mention of the methods, and there is a lot of information about the importance of the results for the needs of water management, which the authors practically did not deal with in the article.

4. The authors defined the goal for the article: "The main objective was to answer the question if the river ice phenomena have been decreasing over the last decades, as it is commonly considered". However, it seems that this hypothesis has been confirmed for Polish and European rivers quite a long time ago.

5. The methods used in the work are indeed quite widely accepted, so it can be concluded that the article is not a novelty in the field of research on ice phenomena.

6. For what purpose was one station with data for the period 1951-2021 analyzed? Is this a representative station for Poland?

7. According to the reviewer, the information in lines 144-148 is unnecessary; they are not relevant to the analysis.

8. The authors adopted the annual number of days with ice as the key variable. How should "with ice" be interpreted - is it any ice phenomenon or a permanent ice cover? This is important because the number of days with ice cover is much smaller than the number of days with all ice phenomena.

9. Note to the description in line 153: Many studies provide the dates of the beginning of ice phenomena on Polish rivers. On what basis did the authors draw such a conclusion? How were these dates verified as incorrect? Was method used to assess the homogeneity of the IMGW-PIB observational series? In this paper, the authors only analyze the number of days - descriptive statistics. The phenomenon exists or does not exist, and the dates of occurrence of ice phenomena have not been studied.

10. The occurrence of ice phenomena on mountain rivers differs from the rest of the country. Climatic conditions - especially air temperature and water temperature, affecting the ice regime of rivers - are different than in the lowland regions and coastal regions of Poland.

Therefore that the title of the work is not adequate to the content of the work. It should be taken into account that the lowlands in Poland cover 91.3% of the area, the highlands 5.6%, and the mountains 3.3%, of which 0.2% is high mountains. The authors write: The selected gauging stations are located mostly in the Carpathian Mountains, in the southern part of Poland, leaving vast areas beyond our investigation".

11. In the _pdf file, page 11 is not filled with content. This is an editorial error, whether some information is missing? - it's hard to guess. Similarly, page 13?

12. Table 2 to modify. I suggest removing the columns for 50 years. and describe Ropa/Klęczany in the text. Empty spaces are not conducive to good reception of information.

13. Note to lines 364-366: There is no basis for such formulations because the authors did not carry out any forecast of changes in the number of ice phenomena on rivers. The obtained results only determine the number of days with ice and determine trends. Trend is not equivalent to forecast!

14. Note to lines 367 - this was not the subject of the authors' research, nor was it the purpose of the authors' research. Only in the Discussion did the authors refer to this issue.

15. Note on line 376: Determining regional patterns requires increasing the number of observations outside the area studied by the authors!

The authors only studied rivers in southern Poland. In the discussion, the authors assume a number of scenarios of continued trends in changes or disappearance of ice phenomena on rivers, referring to the causative factors, but no such analyzes were carried out in the study. The conclusions are only assumptions and are based on the results of previous research conducted in Poland.

16. The discussion does not refer to the results obtained; It's not exactly on the right top.

6. PLOS authors have the option to publish the peer review history of their article (what does this mean?). If published, this will include your full peer review and any attached files.

Reviewer #1: **Yes: **Akram Elentably

Reviewer #2: No

Reviewer #3: No

Reviewer #4: No

Reviewer #5: No

Reviewer #6: **Yes: **SEEMA RANI

Reviewer #7: No

Reviewer #8: No

Reviewer #9: **Yes: **Ting Li

Reviewer #10: No

---

## [Author Response · Author response to Decision Letter 0]

28 May 2024

Dear Editors, dear Reviewers,

We would like to thank very much for the remarks and your valuable time you devoted to review our paper. We believe, that thanks to your remarks the new version, which was significantly reconstructed, will meet the high standards of the PLOS ONE. Since there were ten reviewers, we decided to address the remarks in the tables (attached in the seperate file), which is, in our opinion, the most convenient way of organisation of the remarks and answers.

We count that the new version of the manuscript will satisfy both the Reviewers, Editors and finally the readers.

With best regards

Yours faithfully

Authors

---

## [Decision Letter · Decision Letter 1]

16 Jun 2024

PONE-D-24-05921R1Analysis of changes in the occurrence

of ice phenomena in upland and mountain rivers of PolandPLOS ONE

Dear Dr. Kochanek,

Thank you for submitting your manuscript to PLOS ONE. After careful consideration, we feel that it has merit but does not fully meet PLOS ONE’s publication criteria as it currently stands. Therefore, we invite you to submit a revised version of the manuscript that addresses the points raised during the review process.

We look forward to receiving your revised manuscript.

Kind regards,

Salim Heddam

Academic Editor

PLOS ONE

Journal Requirements:

Additional Editor Comments:

Reviewer 8#:

The article has been largely improved. Not all comments were taken into account.

However, there are few references to Polish literature in the article, which may mean that the authors did not precisely recognize the essence of the problem.

The authors explain that they did not quote M. Grześ's works because it makes it difficult for foreigners to read.

This is in contradiction with the cited works by Paczowska (1938) and Bączyk and Suchożebrski (2012), which are in Polish.

Moreover, the authors quote the article in Italian (see Bonferroni 1936).

Reviewer 9#:

The author highly respects the opinions of the reviewers, and I suggest receiving after after minor revision.

The author has greatly revised and comprehensively improved the article, but there are still the following areas worthy of improvement:

1. The abbreviated NID in line 28 should be placed in front of the word “phenomena”

2. Lines 65 to 82 are the summary of the research area, which is very good, but there is no summary of the common points and differences of the research, just a simple enumeration of the literatures.

The purpose of writing the research background is to extend the research necessity of the author's article, so I hope the author can summarize the article when listing the articles.

3.Line 331 number of days with ice can be abbreviated to NID, and similar problems in other parts of the article are corrected.

4. The full text has an grammatical tense disorder. I suggest to check and revise the full text.

Reviewers' comments:

Reviewer's Responses to Questions

**Comments to the Author**

1. If the authors have adequately addressed your comments raised in a previous round of review and you feel that this manuscript is now acceptable for publication, you may indicate that here to bypass the “Comments to the Author” section, enter your conflict of interest statement in the “Confidential to Editor” section, and submit your "Accept" recommendation.

Reviewer #2: All comments have been addressed

Reviewer #3: All comments have been addressed

Reviewer #4: All comments have been addressed

Reviewer #5: All comments have been addressed

Reviewer #6: All comments have been addressed

Reviewer #7: All comments have been addressed

Reviewer #8: (No Response)

Reviewer #9: All comments have been addressed

Reviewer #10: All comments have been addressed

2. Is the manuscript technically sound, and do the data support the conclusions?

Reviewer #2: Yes

Reviewer #3: Yes

Reviewer #4: Yes

Reviewer #5: Yes

Reviewer #6: Yes

Reviewer #7: Yes

Reviewer #8: Partly

Reviewer #9: Yes

Reviewer #10: Yes

3. Has the statistical analysis been performed appropriately and rigorously? 

Reviewer #2: Yes

Reviewer #3: Yes

Reviewer #4: Yes

Reviewer #5: Yes

Reviewer #6: Yes

Reviewer #7: Yes

Reviewer #8: Yes

Reviewer #9: Yes

Reviewer #10: Yes

4. Have the authors made all data underlying the findings in their manuscript fully available?

Reviewer #2: Yes

Reviewer #3: Yes

Reviewer #4: Yes

Reviewer #5: Yes

Reviewer #6: Yes

Reviewer #7: Yes

Reviewer #8: Yes

Reviewer #9: Yes

Reviewer #10: Yes

5. Is the manuscript presented in an intelligible fashion and written in standard English?

Reviewer #2: Yes

Reviewer #3: Yes

Reviewer #4: Yes

Reviewer #5: Yes

Reviewer #6: Yes

Reviewer #7: Yes

Reviewer #8: Yes

Reviewer #9: No

Reviewer #10: Yes

6. Review Comments to the Author

Reviewer #2: The authors have made all suggested edits to the article. This version of the article may be accepted for publication.

Reviewer #3: Dear authors,

Thank you for addressing my comments satisfactorily and improving your manuscript.

This version is much better.

Regards,

Reviewer #4: The revised manuscript has adequately addressed the concerns raised during the initial review，and the authors have made significant efforts to clarify and improve various sections of the paper, ensuring a higher-quality presentation and analysis.Also，the authors have also improved the quality of the figures, ensuring they are clear and meet publication standards.Therefore, I recommend the manuscript for publication in PLOS ONE.

Reviewer #5: Authors good adressses my comments. It is good job. Paper presents very important analyses in regional scale. I have no new suggestions

Reviewer #6: (No Response)

Reviewer #7: Dear authors, many thanks for your kind revision. It is very much appreciated. Your manuscript can be accepted now.

Reviewer #8: The article has been largely improved. Not all comments were taken into account. However, there are few references to Polish literature in the article, which may mean that the authors did not precisely recognize the essence of the problem. The authors explain that they did not quote M. Grześ's works because it makes it difficult for foreigners to read. This is in contradiction with the cited works by Paczowska (1938) and Bączyk and Suchożebrski (2012), which are in Polish. Moreover, the authors quote the article in Italian (see Bonferroni 1936).

I leave the decision to the editor

Reviewer #9: The author highly respects the opinions of the reviewers, and I suggest receiving after after minor revision. The author has greatly revised and comprehensively improved the article, but there are still the following areas worthy of improvement:

1. The abbreviated NID in line 28 should be placed in front of the word “phenomena”

2. Lines 65 to 82 are the summary of the research area, which is very good, but there is no summary of the common points and differences of the research, just a simple enumeration of the literatures. The purpose of writing the research background is to extend the research necessity of the author's article, so I hope the author can summarize the article when listing the articles.

3.Line 331 number of days with ice can be abbreviated to NID, and similar problems in other parts of the article are corrected.

4. The full text has an grammatical tense disorder. I suggest to check and revise the full text.

Reviewer #10: The manuscript: “Analysis of changes in the occurrence of ice phenomena in upland and mountain rivers of Poland" " is a revised and re-submitted version of a previously manuscript.

Previous title: "Trends in Ice Phenomena on Polish Rivers".

The authors referred to comments in the earlier review. The manuscript was revised thoroughly; all of each section are presented clearly. The authors made a range of modifications and adjustments throughout to improve the precision of the writing and wording, the conclusions are sufficient and informative.

In relation to the first version of the article, the authors have made appropriate changes and additions, which constitute a response to the questions and comments of the reviewer.

The authors:

- changed the title of the manuscript in accordance with the content and database presented therein

- explained in detail some issues related to the methodological approach used in the work - the most important explanations and findings

-modified some paragraphs of the article, thus answering the reviewer's questions and explaining the understatement and complementing lack of explanations.

- corrected the discussion of results and conclusions from the research

In the opinion of the reviewer, the authors' explanations and supplements are appropriate and satisfactory.

7. PLOS authors have the option to publish the peer review history of their article (what does this mean?). If published, this will include your full peer review and any attached files.

Reviewer #2: No

Reviewer #3: No

Reviewer #4: No

Reviewer #5: No

Reviewer #6: **Yes: **SEEMA RANI

Reviewer #7: No

Reviewer #8: No

Reviewer #9: **Yes: **Ting Li

Reviewer #10: No

---

## [Author Response · Author response to Decision Letter 1]

30 Jun 2024

Answers to the Reviewers’ remarks

Dear Editors, dear Reviewers

We would like to thank very much Prof. Seema Rani, Prof. Ting Li and other anonymous Reviewers for the remarks and your valuable time you devoted to review our paper. We believe, that thanks to your remarks the new version will meet the high standards of the PLOS ONE and will be ready for publication. In the following document we concentrate only on the critical remarks answering to them and changing manuscript accordingly. At the same time we express our appreciation for good words about our work.

We count that the new version of the manuscript will satisfy both the Reviewers, Editors and finally the readers.

With best regards

Yours faithfully

Authors

Reviewer #8: The article has been largely improved. Not all comments were taken into account. However, there are few references to Polish literature in the article, which may mean that the authors did not precisely recognize the essence of the problem. The authors explain that they did not quote M. Grześ's works because it makes it difficult for foreigners to read. This is in contradiction with the cited works by Paczowska (1938) and Bączyk and Suchożebrski (2012), which are in Polish. Moreover, the authors quote the article in Italian (see Bonferroni 1936).

Answer: Thank you very much for drawing our attention to the Prof Grześ’ publications. We considered the monumental works by Prof. Grześ in new version of the article.

Reviewer #9: The author highly respects the opinions of the reviewers, and I suggest receiving after after minor revision. The author has greatly revised and comprehensively improved the article, but there are still the following areas worthy of improvement:

1. The abbreviated NID in line 28 should be placed in front of the word “phenomena”

Answer: Thank you very much for this suggestion; we corrected the text accordingly.

2.Lines 65 to 82 are the summary of the research area, which is very good, but there is no summary of the common points and differences of the research, just a simple enumeration of the literatures. The purpose of writing the research background is to extend the research necessity of the author's article, so I hope the author can summarize the article when listing the articles.

Answer: We corrected this fragment according the Reviewer’s suggestions.

3.Line 331 number of days with ice can be abbreviated to NID, and similar problems in other parts of the article are corrected.

Answer: We corrected the manuscript accordingly.

4.The full text has an grammatical tense disorder. I suggest to check and revise the full text.

Answer: We revised the article accordingly. We tried to keep the narration in past tense, unless other tense was necessary to highlight the discussed issue.

---

## [Decision Letter · Decision Letter 2]

12 Jul 2024

Analysis of changes in the occurrence

of ice phenomena in upland and mountain rivers of Poland

PONE-D-24-05921R2

Dear Dr. Kochanek

We’re pleased to inform you that your manuscript has been judged scientifically suitable for publication and will be formally accepted for publication once it meets all outstanding technical requirements.

Kind regards,

Salim Heddam

Academic Editor

PLOS ONE

Additional Editor Comments (optional):

Reviewers' comments:

Reviewer's Responses to Questions

**Comments to the Author**

1. If the authors have adequately addressed your comments raised in a previous round of review and you feel that this manuscript is now acceptable for publication, you may indicate that here to bypass the “Comments to the Author” section, enter your conflict of interest statement in the “Confidential to Editor” section, and submit your "Accept" recommendation.

Reviewer #8: All comments have been addressed

Reviewer #9: All comments have been addressed

2. Is the manuscript technically sound, and do the data support the conclusions?

Reviewer #8: Yes

Reviewer #9: Yes

3. Has the statistical analysis been performed appropriately and rigorously? 

Reviewer #8: I Don't Know

Reviewer #9: Yes

4. Have the authors made all data underlying the findings in their manuscript fully available?

Reviewer #8: No

Reviewer #9: Yes

5. Is the manuscript presented in an intelligible fashion and written in standard English?

Reviewer #8: Yes

Reviewer #9: Yes

6. Review Comments to the Author

Reviewer #8: (No Response)

Reviewer #9: The author has revised the grammar of the entire text to meet publication requirements. Suggest accepting directly.

7. PLOS authors have the option to publish the peer review history of their article (what does this mean?). If published, this will include your full peer review and any attached files.

Reviewer #8: No

Reviewer #9: **Yes: **Ting Li

---

## [Editor Report · Acceptance letter]

18 Jul 2024

PONE-D-24-05921R2 

PLOS ONE

Dear Dr. Kochanek, 

I'm pleased to inform you that your manuscript has been deemed suitable for publication in PLOS ONE. Congratulations! Your manuscript is now being handed over to our production team.

Kind regards, 

on behalf of

Dr. Salim Heddam 

Academic Editor

PLOS ONE